# Natural and Historical Heritage of the Lisbon Botanical Gardens: An Integrative Approach with Tree Collections

**DOI:** 10.3390/plants10071367

**Published:** 2021-07-04

**Authors:** Ana Raquel Cunha, Ana Luísa Soares, Miguel Brilhante, Pedro Arsénio, Teresa Vasconcelos, Dalila Espírito-Santo, Maria Cristina Duarte, Maria Manuel Romeiras

**Affiliations:** 1Linking Landscape, Environment, Agriculture and Food (LEAF), Instituto Superior de Agronomia (ISA), Universidade de Lisboa, Tapada da Ajuda, 1340-017 Lisboa, Portugal; arcunha@isa.ulisboa.pt (A.R.C.); miguelbrilhante131@hotmail.com (M.B.); arseniop@isa.ulisboa.pt (P.A.); dalilaesanto@isa.utl.pt (D.E.-S.); 2Research Centre in Biodiversity and Genetic Resources (CIBIO/InBIO), School of Agriculture, University of Lisbon, Tapada da Ajuda, 1349-017 Lisbon, Portugal; alsoares@isa.ulisboa.pt; 3Jardim Botânico da Ajuda, Instituto Superior de Agronomia (ISA), Universidade de Lisboa, Calçada da Ajuda, 1300-011 Lisboa, Portugal; 4Instituto Superior de Agronomia (ISA), Universidade de Lisboa, Calçada da Ajuda, 1300-011 Lisboa, Portugal; tvasconcelos@isa.ulisboa.pt; 5Centre for Ecology, Evolution and Environmental Changes (cE3c), Faculdade de Ciências, Universidade de Lisboa, Campo Grande, 1749-016 Lisboa, Portugal; mcduarte@fc.ul.pt

**Keywords:** historical heritage, living trees collections, Portugal, spermatophytes, trees, conservation, urban green spaces

## Abstract

Botanical gardens have long contributed to plant science and have played a leading role in ex situ conservation, namely of threatened tree species. Focusing on the three botanical gardens of Lisbon (i.e., Botanical Garden of Ajuda—JBA, Lisbon Botanical Garden—JBL, and Tropical Botanical Garden—JBT), this study aims to reveal their natural heritage and to understand the historical motivations for their creation. Our results showed that these gardens contain a total of 2551 tree specimens, corresponding to 462 taxa, within 80 plant families. Of these, 85 taxa are found in the three gardens, and more than half of the taxa are hosted in JBL (334 taxa), whereas 230 and 201 taxa were recorded in JBT and JBA, respectively. The motivations for the creation of each garden are reflected in the different geographic origins of the trees they host in their living collections. The Palearctic species are dominant in JBA and JBL, and Tropical trees prevail in JBT. With more than 250 years of history, these gardens hold an invaluable natural and historical heritage, with their living collections providing valuable sources of information for the conservation of threatened plant species, at local and global scales.

## 1. Introduction

Biodiversity loss is a global phenomenon [1]. Adding to the conservation of species diversity in their natural habitats, the loss and degradation of urban environments made the preservation of urban ecosystems and the promotion of their biodiversity particularly important since the beginning of the 21st century [2]. Many cities still include remains of previous forests and trees that should be actively managed and preserved, namely in urban parks or woodlots [3]. Moreover, botanical and environmental awareness will be better raised among the community by providing education and places to learn about the plants and their conservation, and the botanical gardens play a relevant role in this context [4,5].

Botanical gardens are institutions that keep well-documented collections of living plants and seed banks, and are thus fundamental for conservation, education, and science [6,7]. Since the 1970s, the conservation of species outside their natural habitat (ex situ conservation), namely of threatened tree species, is encouraged by the International Union for Conservation of Nature (IUCN) and Botanic Gardens Conservation International (BGCI) [8]. As unique urban green spaces, the botanical gardens offer the potential to restore populations of threatened species, as their living collections and seed banks are insurance against the extinction of species in their natural habitats and provide means to reintroduce or reinforce their populations thereby making their survival possible [9,10].

In plant conservation, trees are widely recognized as critically important to the planet and population at the economic, cultural, and ecological levels [7]. However, their numbers have been declining in situ due to, e.g., climate change, overgrazing, agriculture, and logging [11]. According to the IUCN Red List, ca. 10,000 tree species are threatened with extinction worldwide [12]. Several studies have also recognized that trees are key elements in urban nature, as their presence provides numerous services to the urban ecosystem in ecological (i.e., climate, air quality, hydrology, soil, and biodiversity conservation), social, cultural, and aesthetic terms [13].

Lisbon, capital of Portugal, has unique characteristics owing to its location in Southwest Europe, in a transition zone between the Atlantic Ocean and the Mediterranean Sea, and between Africa and Eurasia, which allows it to host a higher average biodiversity than other European cities [14]. According to the municipality, Lisbon has about 148 gardens and parks, among which are three botanical gardens: Botanical Garden of Ajuda (hereafter, JBA), Lisbon Botanical Garden (hereafter, JBL), and Tropical Botanical Garden (hereafter, JBT) (see Figure 1), all of them belonging to the University of Lisbon. Lisbon is the only city in Portugal housing three botanical gardens, and they hold a rich natural and historical heritage, a valuable scientific resource to promote interdisciplinary research activities between the different schools of the University, as well as with worldwide institutions interested in the study of plant diversity and conservation. These three botanical gardens are very different, created at different times but with common overall purposes: research, plant conservation, horticulture, and education. Surely, the study of these gardens and of their collections would enhance the current understanding on Garden Art History and Plant Diversity, but the knowledge of this heritage is often incomplete and outdated.

Knowing the botanical and historical heritage of the three Botanical Gardens of Lisbon, and what it represents, is essential to provide the first global measurement of ex situ conservation of living tree collections. Rivers et al. [10] advocate the need to divulge the ex situ collections housed in botanical gardens and arboreta, to plan future collecting efforts and to adopt integrated global approaches to protect threatened tree species.

With the present work, we aimed to contribute to disseminating the heritage represented by the Botanical Gardens of Lisbon, by coupling the history and the characteristics of the tree layer that dominates them. With this approach, we intended to explain why the presence of three Botanical Gardens in Lisbon is not redundant and to highlight their continued relevance for knowledge and for urban sustainability.

## 2. Results

### 2.1. History of the Botanical Gardens of Lisbon

#### 2.1.1. Botanical Garden of Ajuda (JBA)

The first Portuguese botanical garden, *Jardim Botânico da Ajuda* (Botanical Garden of Ajuda, see Figure 1), was installed on the hill slope of Ajuda after the big 1755 earthquake. Founded in 1768, during the reign of José I (1714–1777) by the indication of his state secretary, Sebastião Carvalho e Mello (1699–1782), the garden was created by the Italian naturalist Domingos Vandelli (1735–1816), who was its first director. Along with the garden, the Royal Cabinet of Natural History, a Chemistry Laboratory, the *Casa do Risco*, where designers created scientific illustrations, and a Specialized Library were also created, as well as, after 1777, a cabinet of Experimental Physics.

The Royal Cabinet of Natural History became one of the richest European scientific institutions, as a result of the “*Philosophical Journeys*” to the Portuguese territories overseas, after the 1780s. For this reason, its collection caught great attention in Europe, particularly of naturalists from the *Musée National d’Histoire Naturelle de Paris*. An example of this was the visit of Étienne Geoffroy Saint-Hilaire (1772–1844), in 1807, who studied and selected specimens from JBA to enrich the collection of the *Jardin des Plantes*, in Paris. During the French invasions, part of the collections of the garden and of the Royal Cabinet was taken by the Napoleonic troops to the present *Musée National d’Histoire Naturelle* in Paris. Another part was taken to the Botanical Garden of Rio de Janeiro, when the Portuguese royal family left Lisbon and went to Brazil, in 1807, before the Napoleonic forces invaded Lisbon.

The second director of JBA was Félix da Silva e Avelar Brotero (1744–1828), a famous Portuguese naturalist that Vandelli met in Coimbra, a pioneer in the studies of Botany in Portugal; he wrote the first Portuguese Flora and started the first practical school of Botany of the country at JBA. In 1836, the administration of the garden was assigned to the Academy of Sciences of Lisbon, and its patrimony was once again split up. In January 1837, the Polytechnic School (later, Faculty of Sciences of the University of Lisbon) was created in the center of Lisbon, and its association with a botanical garden was considered indispensable. As a result, some collections of JBA were transferred there in October 1838. Since 1910, the direction of JBA is the responsibility of *Instituto Superior de Agronomia*, University of Lisbon. In 1944, JBA was listed as *Imóvel de Interesse Público* (“Building of Public Interest”) by the Ministry of Culture.

#### 2.1.2. Lisbon Botanical Garden (JBL)

The *Jardim Botânico de Lisboa* (Lisbon Botanical Garden, see Figure 1) was created in 1873, with many collections brought from JBA. Inaugurated in 1878, it is presently part of the National Museum of Natural History and Science, which is responsible for the garden’s management. It was a scientific garden intended to complement teaching and botany research at the former Polytechnic School. The chosen place, Mount Olivete, had already more than two centuries of tradition in the study of botany, with the Jesuit College of Cotovia here established between 1609 and 1759. To install the garden, a draft directive was made in 1843. However, plantations only began in 1873, by the initiative of the Count of Ficalho (1837–1903) and Andrade Corvo (1824–1890), teachers at the Polytechnic School. The collection of the enormous diversity of living plants was first made by the German Edmund Goëze (1838–1929) and the French Jules Daveau (1852–1929), from all the territories under Portuguese rule at the time. Goëze, the first chief gardener, outlined the “*Class*” and Daveau was in charge of the “*Arboretum*”. Between 1892 and 1909, the French Henri Cayeux, the second chief-gardener, made important contributions to the embellishment of this garden, by introducing more species and creating new varieties of plants. That is the case of the introduction of succulent dahlias and the creation of the hybrid *Dombeya × cayeuxii* and of the rose cultivar *“Belle Portugaise*”.

The most significant intervention in the garden occurred in the late 1930s and early 1940s. Under director Ruy Telles Palhinha (1871–1957), the original systematic ordering of the upper terrace of JBL was replaced by ecologically meaningful groups of species. In 2010, JBL was listed as a “National Monument” by the Ministry of Culture.

#### 2.1.3. Tropical Botanical Garden (JBT)

The *Jardim Botânico Tropical* (Tropical Botanical Garden, see Figure 1), initially “Colonial Garden”, was created by royal decree in 1906, under the rule of King Carlos I and by the initiative of Counsellor Manuel António Moreira Júnior, the Navy and Overseas Minister, and aimed to support the teaching of Tropical Agriculture.

The French landscape gardener Henri Navel (1878–1963), who studied at the National School of Horticulture of Versailles and worked as a gardener in some important gardens (e.g., Royal Botanic Gardens, Kew), had the task to prepare the space for its new function as a colonial garden, between 1910 and 1917. The director position was inherently held by the full professor of the “Economic Geography and Colonial Cultures” chair at the *Instituto Superior de Agronomia* (University of Lisbon).

The celebrations of the 800th anniversary of Portugal as an independent state and of the 300 years after the restoration of national independence, which took place in Belém with the 1940s “Exhibition of the Portuguese World”, left their mark on the morphology and identity of JBT, where the exhibitions concerning the Tropics were concentrated. The program of this section aimed to portray Portugal as the third colonial world power. Buildings, some still existing, and temporary pavilions were built, each dedicated to one of the colonies, and recreating African villages.

In 1944, the “Colonial Garden” was merged with the “Colonial Agriculture Museum”, also formally created in 1906, resulting in the *Jardim e Museu Agrícola Colonial* (Agriculture and Colonial Garden and Museum). Since its creation, the garden was dependent on the *Instituto Superior de Agronomia*. In 1951, the name changed to *Jardim e Museu Agrícola do Ultramar* (Overseas Garden and Museum of Agriculture), and in 1974 it became part of the *Instituto de Investigação Científica Tropical* (Tropical Research Institute). In 2006 it received its current name, and in 2007 it was listed as a National Monument by the Ministry of Culture. In August 2015, JBT became part of the University of Lisbon, under the administration of the National Museum of Natural History and Science.

### 2.2. Tree Collections

The tree layers of the three Botanical Gardens of Lisbon contain a total of 2551 specimens corresponding to 462 taxa (i.e., 448 species, nine subspecies, four varieties, and one form). Of the 462 taxa inventoried, 85 are shared by the three gardens, 130 by JBA and JBL, 146 by JBL and JBT, and 111 by JBA and JBT (Figure 2). More than half of the taxa recorded in our study were found in JBL (334 taxa), whilst in JBT and JBA 230 and 201 taxa were recorded, respectively (Table 1; Figure 2). The family, habit, taxonomic classification, and native distribution of each taxon are described in Table 1.

There is no clear relationship between the area (ha) (i.e., green spaces) and the number of species in each botanical garden, as shown in Figure 3. Nevertheless, when comparing JBA with JBL, the number of species tended to increase with the garden area, as there are more varied habitat types in JBL able to accommodate a more diverse range of species. Moreover, species diversity can also reflect the path of each botanical garden over time and socioeconomic issues, taking into account the human and financial resources allocated to garden management. However, the greatest difference was caused by the 1941 hurricane, the strongest one affecting Portugal to date: the trees of the JBA, many of them close to 200 years old, were uprooted from the base, while the trees of the JBL were not affected.

The Eudicotyledons are the more numerous species in the three botanical gardens (76% in JBA; 73% in JBL; 73% in JBT), followed by the Gymnosperms (including Ginkgophyta and Coniferophyta) (10% in JBA; 12% in JBL; 10% in JBT), the Monocotyledons (9% in JBA; 11% in JBL; 13% in JBT), and finally the Magnoliids (6% in JBA; 4% in JBL; 4% in JBT) (Table 1; Figure 3).

A total of 80 plant families are represented, the Fabaceae being the dominant one (39 taxa), followed by the Myrtaceae (36 taxa). In contrast, 33 families (e.g., Casuarinaceae, Ginkgoaceae and Lamiaceae) are represented by one taxon only (Table 1). The most dominant families in JBT and JBL are the Arecaceae (ca. 10% and 8%, respectively) and the Fabaceae (ca. 9% and 6%, respectively), whereas the Fabaceae (11%) and the Myrtaceae (7%) are the main ones in JBA.

The most abundant species is *Washingtonia robusta*, with 106 specimens. In contrast, 171 taxa are represented by only one specimen (e.g., *Abies pinsapo, Brugmansia aurea, Hesperocyparis macnabiana, Pinus torreyana*) (see Table 1).

Among the three gardens, most of the inventoried taxa are trees *sensu stricto* (JBA, 60%; JBL, 62%; JBT, 64%), followed by shrubs (17% of the taxa in JBA, 12% in JBL, 11% in JBT); some taxa grow either as trees or shrubs (13% in JBA, 14% in JBL, 11% in JBT). Rosette trees correspond to 9% in JBA, 11% in JBL, and 13% in JBT; stem-succulent shrubs only represent 1% of the taxa in the three gardens (Table 1).

Among the taxa with an IUCN classification (64.1% of the recorded taxa; the other 166 taxa have no IUCN evaluation) (Table 1), 0.9% are categorized as CR (*Araucaria angustifolia*, *Beaucarnea recurvata*, *Fraxinus pensylvanica*, and *Pinus torreyana*), 3.2% as EN (e.g., *Ginkgo biloba*, *Sequoia sempervirens*, *Sideroxylon mirmulans*), 4.3% as VU (e.g., *Dracaena draco*, *Howea forsteriana*, *Jacaranda mimosifolia*), 2.2% as NT (e.g., *Pistacia vera*, *Washingtonia filifera*, *Zelkova serrata*), 51.5% as LC, 1.7% as DD, and 0.2% as EW (*Brugmansia aurea*, which was eradicated in the wild by indigenous populations due to its toxicity).

In JBA, about 1.0% of the taxa are classified as CR, 1.5% as EN, 4.0% as VU, 2.5% as NT, 51.7% as LC, and 2.5% as DD (Figure 4). In JBT, 0.4% of the taxa are classified as CR, 4.3% as EN, 5.2% as VU, 3.0% as NT, 49.1% as LC, and 2.2% as DD. JBL hosts 1.2% of taxa classified as CR, 3.6% as EN, 3.6% as VU, 2.1% as NT, 50.9% as LC, 2.1% as DD, and 0.3% as EW. A considerable portion of the species (JBA 36.0%, JBT 35.7% and JBL 36.2%) remains unevaluated by IUCN.

Comparing the three Botanical Gardens, 29 tree species stand out in JBL because of their conservation status worldwide, followed by JBT and JBA with 23 and 16 species, respectively (Table 1). Eleven of these taxa only occur in the JBL collection, namely *Brugmansia aurea*, while the other six species are exclusive to JBA and JBT collections.

Regarding the worldwide native distribution, 11 main groups were identified (Table 1), namely: Palearctic (20%) (e.g., *Arbutus unedo, Ceratonia siliqua)*; Neotropical (18%) (e.g., *Persea americana, Schinus terebinthifolia*); Oriental (16%) (e.g., *Ginkgo biloba, Ficus religiosa*); Afrotropical (11%) (e.g., *Persea barbujana, Phoenix canariensis)*; and Australotemperate (11%) (e.g., *Araucaria bidwillii, Brachychiton populneus*). Using the data presented in Table 1, a heatmap was constructed with the vertical columns representing the three botanical gardens and the horizontal lines the biogeographic regions (Figure 5); the resulting combination boxes were blue-colored according to a gradient from the highest number of taxa (darkest blue) to the lowest (white).

The prevailing origin is the Palearctic, in JBA (27%) and JBL (22%), followed by the Neotropical, which corresponds to 16% and 17% of the taxa in both gardens, respectively. In JBT, the main origins are Neotropical (19%) and Oriental (18%), with a considerable number of useful plants such as *Aleurites moluccanus, Cinnamomum burmanni, Casimiroa edulis, Feijoa sellowiana, Eugenia myrcianthes, Persea americana, Psidium cattleyanum* or *Syzygium jambos*. The Afrotemperate, Andean, Australotropical, Neoguinean, and Neozelandic are the least represented regions in the three Gardens (less than 6% of taxa).

#### 2.2.1. Remarkable Species of JBA

At JBA, worth noting are two splendorous *Ficus macrophylla*, and the emblematic dragon tree (*Dracaena draco*) (Figure 6A), probably one of the oldest specimens in the garden (over three centuries old). Other emblematic trees include the only living specimen of *Schotia afra* (Figure 6B) in a European botanical garden [15], *Araucaria bidwillii, Zelkova serrata, Ocotea foetens, Quercus faginea* and *Phytolacca dioica*. On the lower terrace, the boxwood hedges enclose some tree specimens which deserve special attention because of their size or shape, namely *Lagerstroemia indica, Lagunaria patersonia, Dracaena draco*, *Araucaria heterophylla* (Figure 6C), and *Araucaria cunninghamii*. It should be mentioned that *Araucaria* species are majestic trees, full of symbolism in Portuguese gardens.

#### 2.2.2. Remarkable Species of JBL

The collection of plants here is the most diverse among the three gardens. For instance, the outstanding diversity of palms (Figure 7A), brought from all continents, provides unexpected tropical scenarios at several points in the garden [16,17]. This garden holds some collections particularly worth mentioning and some species that stand out because of their conservation status worldwide, namely: *Pinus torreyana, Metasequoia glyptostroboides, Chrysophyllum imperial* (Figure 7B), *Brahea edulis, Afrocarpus mannii, Taxodium distichum* var. *mexicanum* or *Dracaena draco* (Figure 7C), among other species with conservation interest.

#### 2.2.3. Remarkable Species of JBT

Among the most notable specimens in JBT is the collection of rare palm trees, the exuberant *Ficus macrophylla* (one of the largest specimens in Europe) (Figure 8A), and a two-century-old *Yucca gigantea* (Figure 8B). Palms are a dominant element in the garden, as immediately perceived upon entrance, in the main avenue flanked by *Washingtonia robusta* and *Washingtonia filifera*, inspired by the Botanical Garden of Rio de Janeiro (Figure 8C). Additionally remarkable are the *Dracaena draco, Ficus sycomorus* and *Afrocarpus mannii*, among others, e.g., [18].

## 3. Discussion

### 3.1. Natural and Historical Heritage of the Botanical Gardens of Lisbon

The three Botanical Gardens of Lisbon represent an invaluable legacy, and their different scientific and educational roles reflect on their plant collections. Over the centuries, Lisbon’s climate has allowed the coexistence of plant species from many different biogeographical origins, which greatly diversified its gardens. This survey of the tree species (*sensu lato*) present at the three botanical gardens highlighted that the diversity and tree richness of each garden is linked with its historical background. Despite their different ages and histories, the three Botanical Gardens of Lisbon share the same purposes: research, education, and conservation. Their value goes far beyond their roles as simple gardens, as they can contribute to urban forestry and resilient landscapes, to support and advance urban agriculture, and to conserve urban biodiversity [19].

Although the diversity of species might be expected to depend on the area of the garden due to the potentially higher availability of space and/or of varied habitat types [20], this was not clear in our study. In fact, species diversity can also reflect the development of each garden over time, as well as socioeconomic issues, taking the human and financial resources allocated to garden management into account [21]. In the case of the Botanical Gardens of Lisbon, the historical background and, in particular, the specific mission of each garden cannot be disregarded when trying to understand the existing living collections.

The role of JBA was particularly relevant in the late 18th century as a sponsor of the “*Philosophical Journeys*”, the first scientific expeditions to the Portuguese territories overseas [22]. Later, the importance of this garden and of its plant collection decreased in favor of JBL, created in the late 19th century to support teaching at the Polytechnic School.

Standing out as a support for botany learning [23] and well-adjusted to the site and the mild climate of the city, the Lisbon Botanical Garden (JBL) holds the most diverse collection (ca. 334 taxa) of the three studied gardens, in terms of both species and geographic origins (see Figure 5). In the heart of Lisbon, it constitutes a reference in urban biodiversity.

Created in the early 20th century, the Tropical Botanical Garden (JBT) is noteworthy for its exotic and economically valuable plants (e.g., spices, fruits, medicinal, stimulants, and ornamentals) especially from tropical and subtropical regions [18,24], related to the educational role in the study of Tropical Agriculture and the introduction of economic and exotic plants [25]. The importance of JBT in this context was made clear in 1940, when it hosted the “Colonial Section of the Portuguese World Exhibition”. Nowadays it allows the general public contact with the tropical sciences, thus contributing to promoting scientific culture among the Community of Portuguese Speaking Countries [26].

Comparing the diversity of species that occur in the three Botanical Gardens of Lisbon, some trees stand out for their biological and iconic character; such are the cases of *Ginkgo biloba*, the only representative of the Ginkgoaceae family at present [27] and considered a relict taxon dating from the early Jurassic period, a most well-represented species in botanical gardens worldwide [10], and of the Vulnerable *Dracaena draco*, the most emblematic tree of the Macaronesian region, where it is endemic. Its exuberant growth-form (i.e., arborescent and pachycaulous) and its red resin, known as “dragon’s blood”, are of great interest both at the ornamental and ethnobotanical levels, and the species is widely cultivated despite its rarity in the wild, e.g., [28]. The Endangered *Sequoia sempervirens*, native to the Pacific coast of the USA, and the only surviving species of the *Sequoia* genus [29], is found in the collections of JBL and JBT; and the Critically Endangered *Metasequoia glyptostroboides,* found only in JBL, is also a relict taxon, endemic from a very restricted region of central China [30], and the second best-represented tree species in botanical gardens worldwide [10] where it is successfully cultivated, although the environmental requirements in the wild remain poorly known [31].

The historical imprint is clear in the collections of the three Botanical Gardens of Lisbon. For example, 11% of taxa are from the Afrotropical biogeographic region, reflecting the Portuguese colonization of some very floristic-rich African regions (e.g., Angola or Mozambique) [32]. Whereas Palearctic species are dominant in JBA and JBL, the distinctive tropical character prevails in JBT. The numerous fruit trees from South America (namely Brazil), as well as the numerous useful species of Asiatic origin, account for these values. The flora of south-western Indian Ocean territories (Australia, New Guinea and New Zealand) is the least represented, except for the typical presence, in all the gardens, of some Australotemperate elements, including species of the genera *Araucaria, Brachychiton, Eucalyptus, Ficus,* and *Melaleuca.*

The introduction and acclimatization of new exotic plant species in Lisbon, from different geographical regions and with economic and ornamental interest, were not restricted to the botanical gardens. In fact, private gardens also played important roles in the dissemination of such plants, and the taste for these botanic “*novelties*” was transposed to the streets and public gardens of Lisbon. Private gardens, such as the *Parque Monteiro-Mor* and *Tapada das Necessidades* (both from the 18th century), have played a prominent role in the dissemination of such plants through “*art*” and supported by landscape gardeners, horticulturists, and nurseries.

Additionally, the consort King Ferdinand of Saxe-Coburg and Gotha (1816–1885) brought German romanticism to Portugal. With his collecting spirit and botanic taste and a new way of thinking about gardens, he played an important role in the introduction of exotic ornamental plants [33]. The king’s concern for public gardens was constant, offering plants and even the services of his French gardener Bonnard.

Since the middle of 19th century, the interest for “novelties” was spread to street trees and other Lisbon’s public gardens (e.g., *Jardim da Estrela* (1852), *Jardim do Príncipe Real* (1869), *Jardim de S. Pedro de Alcântara* (1864)). Plants such as *Dracaena draco*, *Ginkgo biloba*, *Jacaranda mimosifolia*, or *Araucaria* spp., were exhibited in streets and gardens, and some of them remain today. Particularly, *Araucaria* spp. are majestic trees very appreciated in Portuguese gardens, with the emblematic examples of *A. angustifolia, A. bidwilli, A. cunninghamii,* and *A. heterophylla* standing out. Of these, *A. angustifolia* and *A. heterophylla* are in risk of extinction and are therefore categorized as CR and VU, respectively (see Table 1). Although *A. angustifolia* has been cultivated in the South American rainforests since ancient times [34] and indigenous population use their seeds for food and religious rituals [35], the *Araucaria* forests are compromised due to overexploitation of timber, aggravated by deforestation for agriculture and urbanism. As a consequence, a large part of the native range of *A. angustifolia* was eliminated, and only a residual part remains, about 15% [36].

Botanical gardens make a significant contribution to the ex situ conservation of wild species, and, particularly for the most threatened trees; their presence should be ensured in several ex situ collections, to widen genetic diversity and thereby increase their conservation value [37]. Of the 22 threatened tree species worldwide most frequently represented in ex situ collections [10], nine can be found in the Botanical Gardens of Lisbon, namely: in the three gardens—*Ginkgo biloba* (EN), and *Beaucarnea recurvata* (CR)—in two of the gardens—*Sequoia sempervirens* (EN), and *Cedrus atlantica* (EN)—or only in one of the gardens—*Metasequoia glyptostroboides* (EN), and *Abies pinsapo* (EN). According to Rivers et al. [10], the family with the highest proportion of threatened species in ex situ collections is the Arecaceae (palm trees, 77%). In the Botanical Gardens of Lisbon, the Arecaceae is also the family with the highest number of threatened taxa—*Brahea edulis* (EN), *Butia eriospatha* (VU), *Howea forsteriana* (VU), and *Sabal bermudana* (EN).

Most of these species are tropical species, which might indicate their adaptation and resilience to the Mediterranean climate of Lisbon [14]. However, and although emblematic tropical palms, such as *Washingtonia robusta* (the most dominant palm species overall), are well adapted in the three gardens, it is crucial to monitor those more vulnerable to the pests and diseases which recently reduced the abundance of some palm species [38]. For instance, the Coleopteran *Rhynchophorus ferrugineus* (Olivier) is a pest of *Phoenix dactylifera* and *Phoenix canariensis*, e.g., [39], which are present in all the Botanical Gardens of Lisbon.

### 3.2. Outreach and Education Programs

Located in the strategic and most touristic area of Lisbon (Belém), JBT receives the highest number of visitors among the three Botanical Gardens of Lisbon (140,000 in 2018). In all the three gardens, plant diversity and conservation are the focus of the programs targeting families and general audiences, and they maintain extension and outreach programs, sometimes with the support of the civil society. Guided tours show the botanical collection as well as the historic, artistic, and cultural heritage enclosed in these spaces. Educational activities for school groups and teachers are also proposed, and training courses for touristic guides and gardening courses are regularly organized. Scientific activities are offered year-round, namely in events such as the European Researchers’ Night, sponsored by the European Commission. JBT recently launched a freely available mobile application with interactive maps (e.g., “Trees you must see”, “Garden with History”, “Birds”, and “Biosensors”) offering augmented reality experiences to the visitors. JBL offers twilight visits, whereas JBA promotes cultural activities such as theatre for children. The three Botanical Gardens of Lisbon are part of the “European Route of Historic Gardens”, certified as one of the forty “Cultural Routes of the Council of Europe”.

The future of the botanical gardens, as spaces of knowledge about the plant world, will certainly depend on the reinforcement of research, education and cultural activities. This, in turn, requires the continuous and careful maintenance of the botanical heritage of these living museums. Accordingly, the three Botanical Gardens of Lisbon are expected to: (i) enhance the visibility of their collections and heritage, and to strengthen bonds, partnerships and joint work between different collections; (ii) manage biological collections using modern software; (iii) create a DNA Bank associated to the collections; (iv) disseminate results to the scientific community, and promote innovative education actions and seminars; and (v) promote ex situ conservation initiatives, including maintenance of the seed bank and identification of living collections of rare and threatened species, particularly from the Portuguese flora.

### 3.3. Final Remarks

All the green areas of Lisbon constitute a very important heritage for our ex situ plant diversity conservation, however, the Botanical Gardens of Lisbon must be seen as unique spaces for tree conservation and their articulation with the surroundings, in ecological, aesthetic, cultural, historical, social, and economic terms, is fundamental to the urbanized space of Lisbon [14].

The preservation of trees, as well as of green urban areas which encompass the botanical gardens, is a current purpose aiming at making cities more resilient to climate change [40,41]. This was one of the main objectives of the “European Green Capital 2020” for Lisbon [42], promoting sustainability, biodiversity, and preservation of threatened species; it should also be a common goal to all cities, since more than half of the world’s population currently lives in cities and predictions indicate that more than two thirds will live there by 2050 [43].

In this context, botanical gardens play a very important role in urban sustainability: adding to botanical knowledge, they also store years of practice in horticulture and arboriculture, which is extremely useful to improve urban green spaces in fields such as tree selection and planting, urban forest management plans and restoration [4]. Furthermore, this kind of knowledge can help in the implementation of sustainable environmental management practices, urban biodiversity [44], and the maintenance of urban trees [45], as well as to promote ecosystem services, preserving and valuing biodiversity [46].

## 4. Materials and Methods

### 4.1. Studied Areas

The three Botanical Gardens of Lisbon (JBA—Botanical Garden of Ajuda; JBL—Lisbon Botanical Garden; and JBT—Tropical Botanical Garden) are characterized in Table 2, and their location in Lisbon (see Figure 1).

#### 4.1.1. Botanical Garden of Ajuda (JBA)

JBA is located in Calçada da Ajuda (Figure 1) opposite to the Palácio Nacional da Ajuda and is the oldest Botanical Garden in Portugal. It was established on two terraces, separated by a limestone balustrade, and connected by a central and two lateral staircases authored by the architect Manuel Caetano de Sousa (1730–1802). The upper terrace hosts the botanical collection, and the lower terrace is composed of a central fountain and 4 km of hedges of boxwood and myrtle, in geometric forms around basins and sculptures from the stone’s school of Machado de Castro (1731–1822), possibly conceived as a recreation space for the royal family. JBA combines several styles and epochs, with a predominant line of baroque influence [47].

#### 4.1.2. Lisbon Botanical Garden (JBL)

JBL is located in the center of Lisbon (Figure 1) and occupies the core of the block bordered by the main avenue (*Avenida da Liberdade*) and a large garden square (*Jardim do Príncipe Real*). It is structured in two parts, the “Class” and the “Arboretum” [48,49]. The “Class” develops at the same level as the main building of the National Museum of Natural History and Science/University of Lisbon; the “Arboretum” extends downwards the slope, as an organic composition of flowerbeds, streams, waterfalls, lakes, and passages, surrounded by a pathway [50].

#### 4.1.3. Tropical Botanical Garden (JBT)

JBT is situated in the monumental area of Lisbon, next to *Mosteiro dos Jerónimos* and to the *Palácio de Belém* and covers 800 m along a south-facing hill slope, overlooking the Tagus River (Figure 1). Favored by a privileged microclimate, JBT has plenty of water and has been enriched with a great diversity of exotic plant species. JBT encloses historical buildings such as the Palace of Condes da Calheta (17th century), and marble sculptures of different aesthetic concepts that span over several historical periods, from the 17th to the 20th centuries [18].

### 4.2. Historical Data

The historic framework of the three botanical gardens was retrieved from several types of documents, such as the descriptions of the scientific expeditions performed in the late 18th century in the Portuguese territories overseas, the so-called “*Philosophical Journeys*” [22,49], and other historical documents with descriptions and information referring to their history, e.g., [18,23,24,45,47,48,49,51,52,53,54,55,56,57,58,59,60,61].

### 4.3. Tree Layer Inventory

The surveys in the three botanical gardens focused on tree specimens and were carried out in the framework of the “LX GARDENS” research project (2014–2017, targeting Lisbon’s historic gardens). The following methodology was used for the botanical study: inventory, location (with geographic information systems) and specimen identification. All data were recorded in a relational database built on a SQL server. Location data recorded included the following items: (1) Specimen ID number; (2) Garden code; (3) Species code; (4) Family; (5) Species; (6) Species classifier; (7) Geographic origin of the taxon; (8) Naturality Status in Portugal (i.e., native, non-native and/or invasive); (9) Growth form (meaning the plant’s physiognomy); (10) ETRS 1989 coordinates; (11) Extinction risk assessment using the IUCN Red List [12]. The data of the three botanical gardens were continuously updated until May 2021.

Tree species were chosen for the study due to their perenniality and longevity, which seemed appropriate and important traits for the intended historical approach. Tree definition followed the one proposed by the IUCN’s Global Tree Specialist Group (GTSG): “a woody plant with usually a single stem growing to a height of at least 2 m, or if multi-stemmed, then at least one vertical stem 5 cm in diameter at breast height”. Several habit types were considered: trees *sensu stricto*, shrubs, rosette trees, and stem-succulents. For simplicity, all these types are here considered, in a broad sense, as “trees”, and belonging to the “tree layer” of the gardens. To support species selection, the GlobalTreeSearch [15] was also consulted.

Species identification or validation was made using specialized bibliography, e.g., [62,63,64,65,66,67,68,69,70,71,72] and specimens housed at the *João Carvalho e Vasconcellos herbarium* (LISI)/*Instituto Superior de Agronomia* (University of Lisbon). Scientific names and families mainly follow Plants of the World Online [73] and the World Flora online [74].

### 4.4. Database of the Tree Layers

Plant data of each Lisbon botanical garden, including species names, families, growth form, and native distribution, are summarized in Table 1. The native distribution follows Morrone’s [75] Biogeographical Regions (i.e., Afrotemperate, Afrotropical, Andean, Australotemperate, Australotropical, Nearctic, Neoguinean, Neotropical, Neozelandic, Oriental, and Palearctic) and online databases, namely the Plants of the World Online [73] and the GBIF platform [76]. The conservation status of each species was retrieved from the IUCN Red List of Threatened Species [12].

### 4.5. Data Treatment

All analyses were carried out in the RStudio program version 1.1.456 [77]. In order to detect the diversity patterns of the tree layers of the three Botanical Gardens of Lisbon, the following analyses were performed: scatter pie plot, Euler diagram, and heatmap. All the plots were visualized by ggplot2 [78].

## Figures and Tables

**Figure 1 plants-10-01367-f001:**
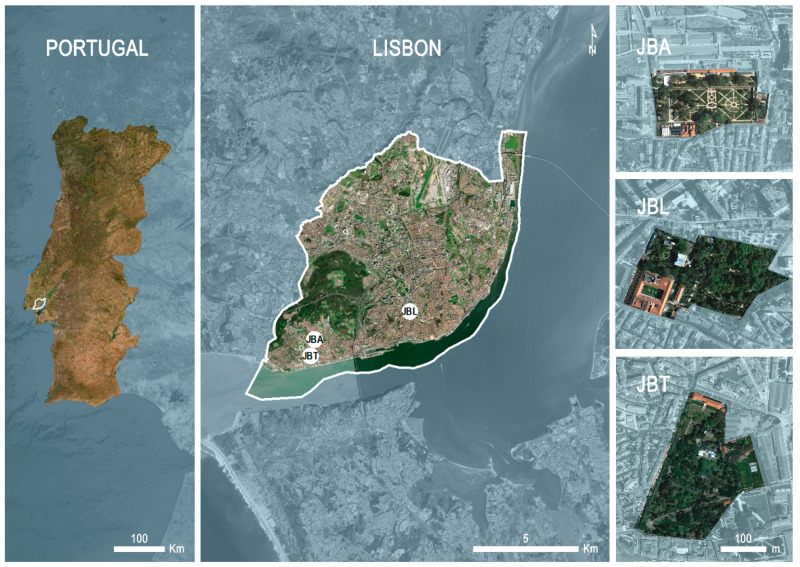
Location of the three Botanical Gardens in Lisbon. Right-top: Botanical Garden of Ajuda (JBA); right-center: Lisbon Botanical Garden (JBL); and right-bottom: Tropical Botanical Garden (JBT).

**Figure 2 plants-10-01367-f002:**
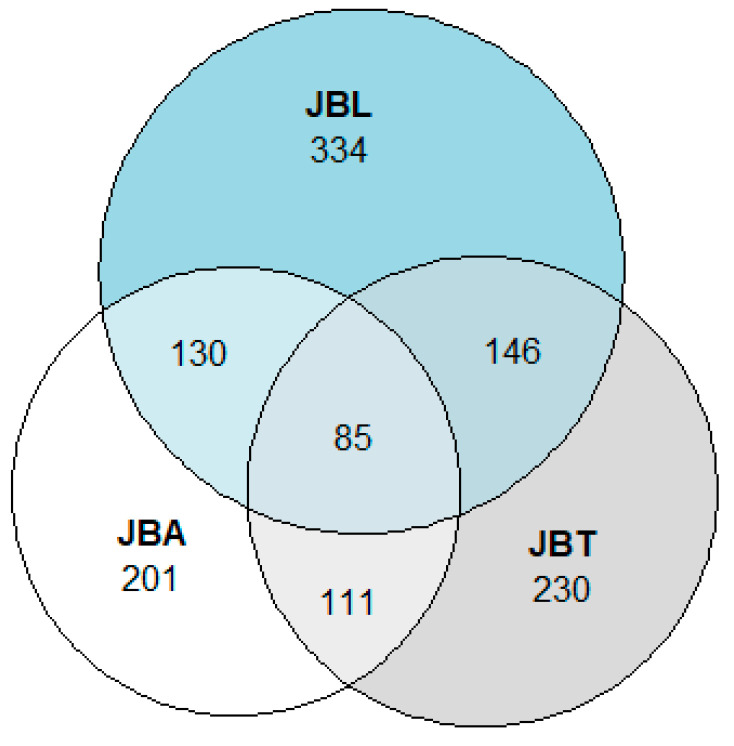
Euler diagram showing the number of taxa of the three Botanical Gardens of Lisbon (JBA, JBL, JBT). The overlapping shapes represent taxa existing in two or three of the gardens.

**Figure 3 plants-10-01367-f003:**
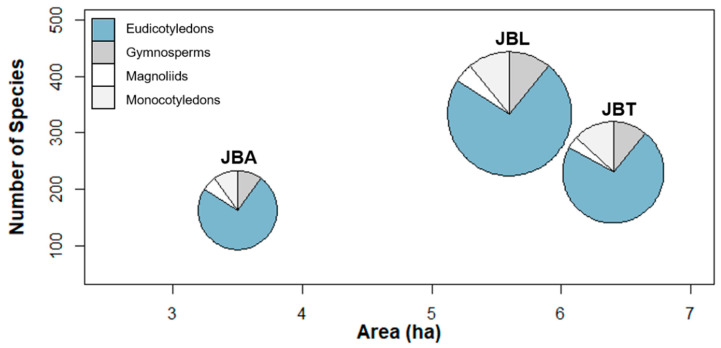
Relationship between garden green area and species richness, in each Botanical Garden of Lisbon (JBA, JBL, JBT); the area of each pie chart represents the number of specimens inventoried for the dominant taxonomic groups that occur in each garden.

**Figure 4 plants-10-01367-f004:**
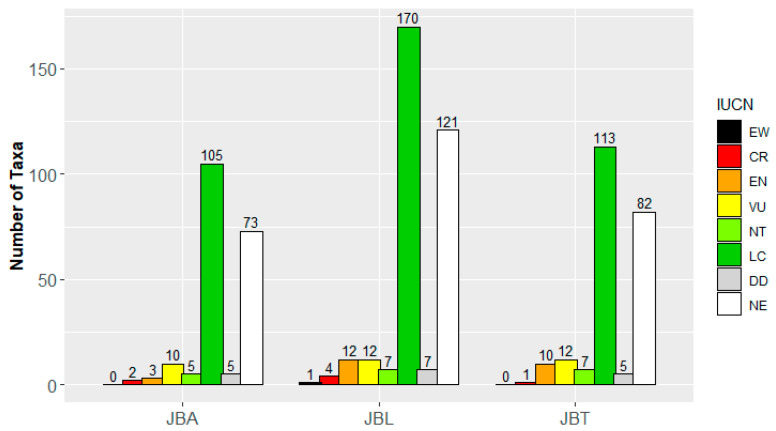
IUCN classification of the taxa composing the arboreal stratum of the three Botanical Gardens of Lisbon. [EW, Extinct in Wild; CR, Critically Endangered; EN, Endangered; VU, Vulnerable; NT, Near Threatened; LC, Least Concern; DD, Data Deficient; NE, Not Evaluated]. The number of species belonging to each IUCN category is shown.

**Figure 5 plants-10-01367-f005:**
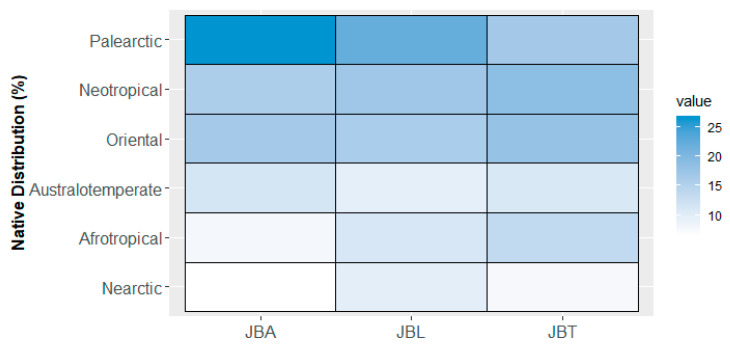
Heatmap showing the biogeographic distribution of taxa found in each Botanical Garden of Lisbon (JBA, JBL, and JBT). Color intensity indicates the relative proportion of the taxa from the six most represented biogeographic origins.

**Figure 6 plants-10-01367-f006:**
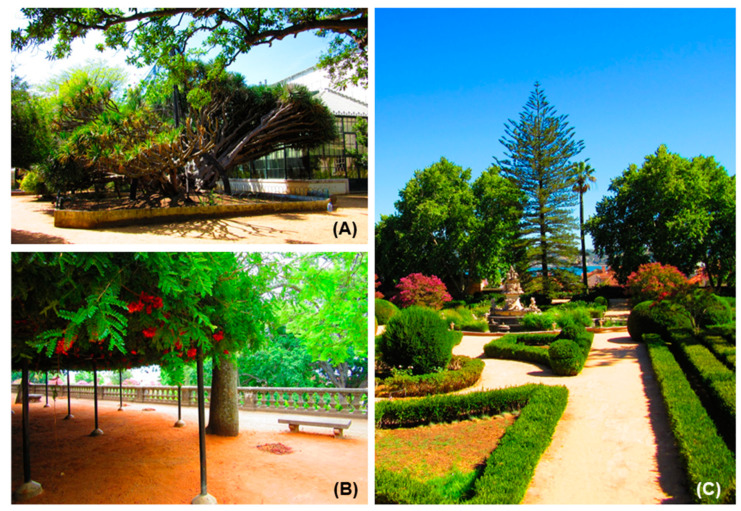
The iconic specimens of JBA: (**A**) *Dracaena draco,* probably one of the oldest specimens in the garden; (**B**) *Schotia afra,* the only living specimen in a European botanical garden; (**C**) Garden perspective showing the specimen of *Araucaria heterophylla* in the background (Photos: Cunha).

**Figure 7 plants-10-01367-f007:**
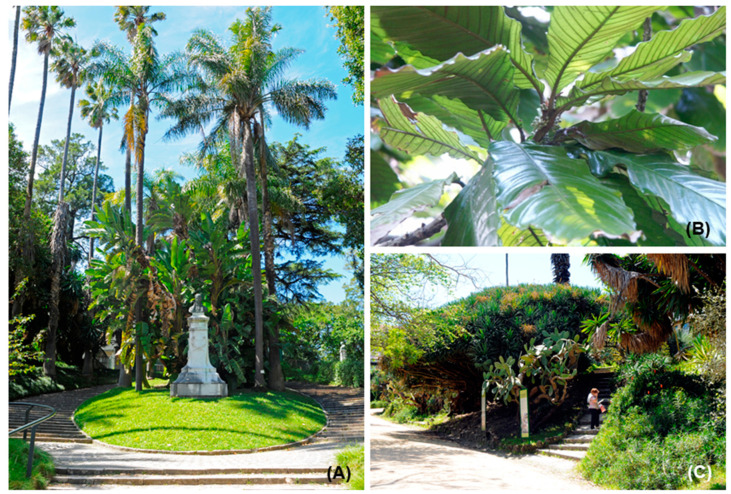
The iconic specimens of JBL: (**A**) Garden perspective showing the outstanding diversity of palms; (**B**) *Chrysophyllum imperiale;* and (**C**) *Dracaena draco* (Photos **A** and **C**: Sachetti; Photo **B**: Forte).

**Figure 8 plants-10-01367-f008:**
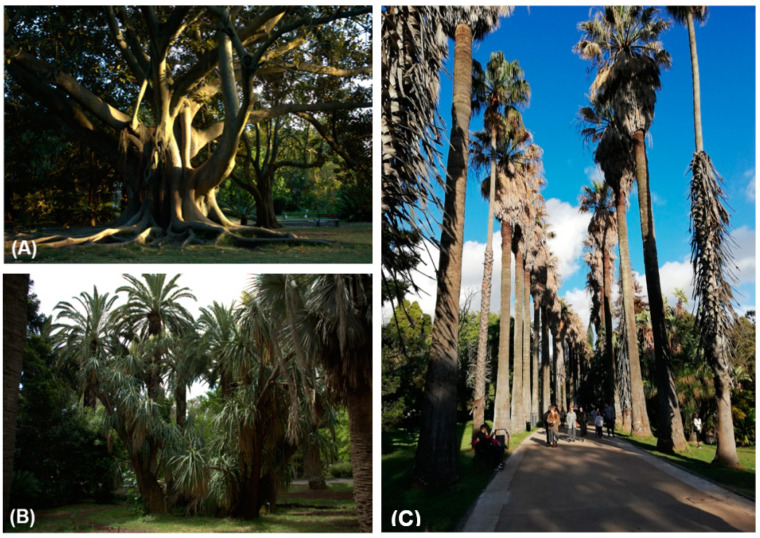
The iconic specimens of JBT: (**A**) *Ficus macrophylla*, one of the most exuberant specimens in the garden and in Europe; (**B**) *Yucca gigantea*, one of the JBT oldest specimens; and (**C**) the main avenue flanked by *Washingtonia robusta* and *Washingtonia filifera*, inspired by the Botanical Garden of Rio de Janeiro (Photos **A** and **B**: Duarte; Photo **C**: Cunha).

**Table 1 plants-10-01367-t001:** Taxa found in the arboreal stratum of the three Botanical Gardens of Lisbon (JBA—Botanical Garden of Ajuda; JBL—Lisbon Botanical Garden; and JBT—Tropical Botanical Garden) in 2021: family, scientific name, occurrence, habit and native distribution.

Family/Taxa	JBA	JBL	JBT	Habit	IUCN ^a^	Native Distribution
Anacardiaceae						
*Harpephyllum caffrum* Bernh.		X		Tree	LC	Afrotropical—Afrotemperate
*Pistacia lentiscus* L.	X	X		Shrub or Tree	LC	Palearctic
*Pistacia vera* L.			X	Tree	NT	Palearctic
*Pleiogynium timoriense* (A.DC.) Leenh.		X		Tree	LC	Australotemperate—Oriental
*Schinus latifolia* (Gillies ex Lindl.) Engl.		X		Tree	NE	Neotropical
*Schinus molle* L.			X	Tree	NE	Andean—Neotropical
*Schinus terebinthifolia* Raddi	X	X	X	Tree	NE	Neotropical
*Spondias mombin* L.		X		Tree	LC	Andean—Neotropical
Annonaceae						
*Annona cherimola* Mill.		X	X	Tree	LC	Andean
Apiaceae						
*Heteromorpha arborescens* var. *abyssinica* (Hochst. ex A. Rich.) H. Wolff		X		Tree	NE	Afrotropical—Afrotemperate
Apocynaceae						
*Acokanthera oblongifolia* (Hochst.) Benth. and Hook.f. ex B.D.Jacks.		X		Tree	LC	Afrotropical—Afrotemperate
*Acokanthera oppositifolia* (Lam.) Codd.			X	Tree	LC	Afrotropical—Afrotemperate
*Carissa bispinosa* (L.) Desf. ex Brenan		X		Shrub	LC	Afrotropical—Afrotemperate
*Carissa macrocarpa* (Eckl.) A.DC.	X	X		Shrub	LC	Afrotropical—Afrotemperate
*Cascabela thevetia* (L.) Lippold	X	X	X	Shrub	LC	Nearctic—Neotropical
*Nerium oleander* L.	X	X	X	Shrub	LC	Afrotropical—Oriental—Palearctic
*Plumeria alba* L.	X		X	Shrub	NE	Neotropical
Aquifoliaceae						
*Ilex aquifolium* L.	X	X		Tree	LC	Palearctic
*Ilex perado* Aiton subsp. *azorica* Tutin		X		Tree	NE	Palearctic
Araliaceae						
*Oreopanax nymphaeifolius* (Hibberd) Decne. and Planch. ex G.Nicholson		X	X	Shrub or Tree	NE	Neotropical
*Plerandra elegantissima* (H.J.Veitch ex Mast.) Lowry, G.M.Plunkett and Frodin	X	X		Tree	EN	Neoguinean
*Pseudopanax lessonii* (DC.) K.Koch		X		Tree	NE	Neozelandic
*Schefflera actinophylla* (Endl.) Harms		X		Tree	LC	Australotropical—Oriental
*Schefflera venulosa* (Wight and Arn.) Harms		X		Shrub or Tree	NE	Oriental
*Tetrapanax papyrifer* (Hook.) K.Koch		X		Shrub	LC	Oriental
Araucariaceae						
*Agathis robusta* (C.Moore ex F.Muell.) F.M.Bailey	X	X	X	Tree	LC	Australotemperate—Neoguinean
*Araucaria angustifolia* (Bertol.) Kuntze		X		Tree	CR	Neotropical
*Araucaria bidwillii* Hook.	X	X	X	Tree	LC	Australotemperate
*Araucaria columnaris* (G.Forst.) Hook.		X	X	Tree	LC	Neoguinean
*Araucaria cunninghamii* Mudie	X	X	X	Tree	LC	Australotemperate—Neoguinean
*Araucaria heterophylla* (Salisb.) Franco	X	X	X	Tree	VU	Neozelandic
Arecaceae						
*Archontophoenix cunninghamiana* (H.Wendl.) H.Wendl. and Drude		X		Rosette tree	NE	Australotemperate
*Arenga pinnata* (Wurmb) Merr.		X		Rosette tree	NE	Oriental
*Bismarckia nobilis* Hildebrandt and H.Wendl.			X	Rosette tree	LC	Afrotropical
*Brahea armata* S.Watson		X	X	Rosette tree	LC	Neotropical
*Brahea edulis* H.Wendl. ex S.Watson		X	X	Rosette tree	EN	Neotropical
*Butia capitata* (Mart.) Becc.		X	X	Rosette tree	NE	Neotropical
*Butia eriospatha* (Mart. ex Drude) Becc.		X		Rosette tree	VU	Neotropical
*Chamaedorea pochutlensis* Liebm.			X	Rosette tree	LC	Nearctic
*Chamaedorea tepejilote* Liebm.			X	Rosette tree	NE	Neotropical
*Chamaerops humilis* L.	X	X	X	Rosette tree	LC	Palearctic
*Howea forsteriana* (F.Muell.) Becc.	X	X	X	Rosette tree	VU	Australotemperate
*Livistona australis* (R.Br.) Mart.		X	X	Rosette tree	NE	Australotemperate
*Livistona chinensis* (Jacq.) R.Br. ex Mart.		X	X	Rosette tree	NE	Oriental
*Phoenix canariensis* H.Wildpret	X	X	X	Rosette tree	LC	Afrotropical
*Phoenix dactylifera* L.	X	X	X	Rosette tree	NE	Afrotropical—Palearctic
*Phoenix loureiroi* Kunth		X	X	Rosette tree	LC	Oriental
*Phoenix reclinata* Jacq.		X	X	Rosette tree	LC	Afrotropical
*Phoenix rupicola* T.Anderson		X		Rosette tree	NT	Oriental
*Rhapis excelsa* (Thunb.) Henry	X	X	X	Rosette tree	NE	Oriental
*Rhopalostylis baueri* (Hook.f.) H.Wendl. and Drude		X		Rosette tree	NE	Australotemperate
*Rhopalostylis sapida* (Sol. ex G.Forst.) H.Wendl. and Drude		X		Rosette tree	NE	Neozelandic
*Sabal bermudana* L.H. Bailey		X	X	Rosette tree	EN	Neotropical
*Sabal minor* (Jacq.) Pers.		X	X	Rosette tree	LC	Nearctic—Neotropical
*Sabal palmetto* (Walter) Lodd. ex Schult and Schult.f.		X	X	Rosette tree	NE	Nearctic—Neotropical
*Serenoa repens* (W.Bartram) Small		X		Rosette tree	NE	Nearctic
*Syagrus romanzoffiana* (Cham.) Glassman		X	X	Rosette tree	NE	Neotropical
*Trachycarpus fortunei* (Hook.) H.Wendl.	X	X	X	Rosette tree	NE	Oriental
*Trachycarpus martianus* (Wall. ex Mart.) H.Wendl.		X		Rosette tree	NE	Oriental
*Trithrinax brasiliensis* Mart.		X		Rosette tree	DD	Neotropical
*Washingtonia filifera* (Rafarin) H.Wendl. ex de Bary	X	X	X	Rosette tree	NT	Nearctic—Neotropical
*Washingtonia robusta* H.Wendl.	X	X	X	Rosette tree	NE	Neotropical
Asparagaceae						
*Beaucarnea recurvata* (K.Koch and Fintelm.) Lem.	X	X	X	Rosette tree	CR	Neotropical
*Beaucarnea stricta* (K.Koch and Fintelm.) Lem.	X			Rosette tree	VU	Neotropical
*Cordyline australis* (G.Forst.) Endl.	X	X	X	Rosette tree	NE	Neozelandic
*Cordyline indivisa* (G. Forst.) Endl.	X			Rosette tree	LC	Neozelandic
*Dasylirion wheeleri* S.Watson ex Rothr.		X		Rosette tree	LC	Neotropical
*Dracaena draco* (L.) L.	X	X	X	Rosette tree	VU	Afrotropical
*Nolina parviflora* (Kunth) Hemsl.		X		Rosette tree	NE	Neotropical
*Yucca aloifolia* L.	X	X	X	Rosette tree	NE	Nearctic—Neotropical
*Yucca carnerosana* (Trel.) McKelvey		X		Rosette tree	LC	Neotropical
*Yucca gigantea* Lem.	X	X	X	Rosette tree	NE	Neotropical
*Yucca gloriosa* L.	X	X		Rosette tree	LC	Nearctic
*Yucca treculeana* Carrière			X	Rosette tree	NE	Nearctic—Neotropical
Asphodelaceae						
*Aloidendron barberae* (Dyer) Klopper and Gideon F.Sm.	X		X	Rosette tree	NE	Afrotropical—Afrotemperate
Asteraceae						
*Montanoa bipinnatifida* (Kunth) K.Koch	X	X	X	Shrub or Tree	NE	Neotropical
*Podachaenium eminens* (Lag.) Sch.Bip. ex Sch.Bip.		X		Shrub	LC	Neotropical
Berberidaceae						
*Berberis bealei* Fortune	X		X	Shrub	NE	Oriental
*Berberis japonica* (Thunb.) Spreng.		X		Shrub	NE	Palearctic
Betulaceae						
*Alnus cordata* (Loisel.) Duby		X		Tree	LC	Palearctic
*Betula pubescens* Ehrh.		X		Tree	LC	Palearctic
*Corylus avellana* L.	X	X		Tree	LC	Palearctic
*Ostrya carpinifolia* Scop.		X		Tree	LC	Palearctic
Bignoniaceae						
*Catalpa bignonioides* Walter	X	X	X	Tree	DD	Nearctic
*Jacaranda mimosifolia* D.Don	X	X	X	Tree	VU	Neotropical
*Markhamia lutea* (Benth.) K. Schum.			X	Tree	LC	Afrotropical
*Radermachera sinica* (Hance) Hemsl.		X		Tree	LC	Oriental—Palearctic
*Spathodea campanulata* P.Beauv.			X	Tree	LC	Afrotropical
Boraginaceae						
*Ehretia acuminata* R.Br.		X		Tree	LC	Australotemperate—Oriental—Neoguinean
*Ehretia anacua* (Terán and Berland.) I.M.Johnst.		X		Tree	LC	Neotropical
Cactaceae						
*Cereus hildmannianus* K.Schum.		X		Stem-succulent shrub	LC	Neotropical
*Cereus hildmannianus* subsp. *uruguayanus* (F.Ritter ex R.Kiesling) N.P.Taylor		X	X	Stem-succulent shrub	NE	Neotropical
*Opuntia leucotricha* DC.	X	X		Stem-succulent shrub	LC	Neotropical
*Opuntia monacantha* Haw.			X	Stem-succulent shrub	LC	Neotropical
*Rhodocactus grandifolius* (Haw.) F.M.Knuth		X		Stem-succulent shrub	LC	Neotropical
Calycanthaceae						
*Chimonanthus praecox* (L.) Link	X	X		Shrub or Tree	LC	Oriental
Cannabaceae						
*Celtis australis* L. subsp. *australis*	X	X	X	Tree	LC	Palearctic
*Celtis caucasica* Willd.		X		Tree	LC	Palearctic
*Celtis occidentalis* L.		X		Tree	LC	Nearctic
*Celtis reticulata* Torr.		X		Tree	NE	Nearctic
*Celtis sinensis* Pers.	X	X		Tree	LC	Oriental
Casuarinaceae						
*Casuarina cunninghamiana* Miq.	X	X	X	Tree	NE	Australotemperate—Australotropical
*Elaeodendron papillosum* Hochst.		X		Tree	NE	Afrotropical—Afrotemperate
*Euonymus europaeus* L.	X	X		Shrub	NE	Palearctic
*Euonymus japonicus* Thunb.		X	X	Shrub	VU	Palearctic
*Maurocenia frangula* Mill		X		Tree	NE	Afrotemperate
Cercidiphyllaceae						
*Cercidiphyllum magnificum* (Nakai) Nakai		X		Tree	NE	Palearctic
Combretaceae						
*Terminalia australis* Cambess.		X		Tree	LC	Neotropical
Cornaceae						
*Cornus capitata* Wall.		X		Tree	LC	Oriental
*Cornus walteri* Wangerin		X		Tree	LC	Oriental
Corynocarpaceae						
*Corynocarpus laevigatus* J.R.Forst. and G.Forst.	X	X	X	Tree	NE	Neozelandic
Cupressaceae						
*Chamaecyparis lawsoniana* (A.Murray bis) Parl.	X	X	X	Tree	NT	Nearctic
*Cupressus sempervirens* L.	X	X	X	Tree	LC	Palearctic
*Hesperocyparis glabra* (Sudw.) Bartel			X	Tree	LC	Nearctic
*Hesperocyparis lusitanica* (Mill.) Bartel	X	X	X	Tree	LC	Neotropical
*Hesperocyparis macnabiana* (A.Murray bis) Bartel			X	Tree	VU	Nearctic
*Hesperocyparis macrocarpa* (Hartw.) Bartel	X	X	X	Tree	LC	Nearctic
*Juniperus cedrus* Webb and Berthel.			X	Tree	EN	Afrotropical
*Juniperus chinensis* L.	X	X		Tree	LC	Oriental—Palearctic
*Juniperus phoenicea* L.	X			Tree	LC	Palearctic
*Juniperus sabina* L.	X			Tree	LC	Palearctic
*Juniperus virginiana* L.		X		Tree	LC	Nearctic
*Metasequoia glyptostroboides* Hu and W.C.Cheng		X		Tree	EN	Oriental
*Platycladus orientalis* (L.) Franco	X	X	X	Tree	NT	Palearctic
*Sequoia sempervirens* (D.Don) Endl.		X	X	Tree	EN	Nearctic
*Taxodium distichum* (L.) Rich.		X		Tree	LC	Nearctic
*Taxodium distichum* var. *mexicanum* (Carrière) Gordon and Glend.		X		Tree	NE	Neotropical
*Tetraclinis articulata* (Vahl) Mast.	X			Tree	LC	Palearctic
*Thuja occidentalis* L.	X	X	X	Shrub or Tree	LC	Nearctic
*Thuja plicata* Donn ex D.Don	X	X		Tree	LC	Nearctic
Curtisiaceae						
*Curtisia dentata* (Burm.f.) C.A.Sm		X		Shrub or Tree	LC	Afrotropical—Afrotemperate
Didiereaceae						
*Portulacaria afra* Jacq.	X	X	X	Shrub	LC	Afrotropical—Afrotemperate
Ebenaceae						
*Diospyros kaki* L.f.	X		X	Tree	NE	Oriental—Palearctic
*Diospyros lotus* L.		X		Tree	LC	Palearctic
*Diospyros virginiana* L.			X	Tree	NE	Nearctic
Elaeagnaceae						
*Elaeagnus angustifolia* L.	X			Shrub	LC	Palearctic
*Elaeagnus pungens* Thunb.	X			Shrub	LC	Oriental
Ericaceae						
*Arbutus unedo* L.	X	X		Shrub or Tree	LC	Palearctic
Eucommiaceae						
*Eucommia ulmoides* Oliv.			X	Tree	VU	Oriental
Euphorbiaceae						
*Alchornea cordifolia* (Schumach. and Thonn.) Müll.Arg.			X	Shrub or Tree	LC	Afrotropical
*Aleurites moluccanus* (L.) Willd.			X	Tree	LC	Oriental
*Baloghia inophylla* (G.Forst.) P.S.Green		X		Tree	NE	Australotemperate
*Euphorbia ingens* E.Mey. ex Boiss.	X		X	Shrub	LC	Afrotropical
*Euphorbia pulcherrima* Willd. ex Klotzsch.	X		X	Shrub	LC	Neotropical
*Euphorbia tirucalli* L.	X		X	Shrub	LC	Afrotropical—Afrotemperate—Oriental
*Euphorbia triangularis* Desf. ex A. Berger		X	X	Shrub	LC	Afrotropical—Afrotemperate
*Mallotus japonicus* (L.f.) Müll.Arg.	X	X		Tree	NE	Palearctic
*Ricinus communis* L.		X	X	Shrub	NE	Afrotropical
Fabaceae						
*Albizia julibrissin* Durazz.	X			Tree	NE	Oriental—Palearctic
*Bauhinia acuminata* L.	X		X	Shrub or Tree	LC	Australotropical—Oriental
*Bauhinia forficata* Link		X	X	Shrub or Tree	LC	Neotropical
*Bauhinia purpurea* L.			X	Shrub or Tree	LC	Oriental
*Bauhinia variegata* L.	X		X	Shrub or Tree	LC	Oriental
*Calliandra tweediei* Benth.	X			Shrub or Tree	LC	Neotropical
*Castanospermum australe* A.Cunn. ex Mudie	X			Tree	NE	Australotemperate
*Ceratonia siliqua* L.	X	X		Tree	LC	Palearctic
*Cercis siliquastrum* L.	X	X	X	Tree	LC	Palearctic
*Dermatophyllum secundiflorum* (Ortega) Gandhi and Reveal		X		Shrub or Tree	NE	Neotropical
*Erythrina americana* Mill.	X		X	Tree	NE	Neotropical
*Erythrina caffra* Thunb.		X	X	Tree	LC	Afrotropical—Afrotemperate
*Erythrina crista-galli* L.	X	X		Tree	LC	Neotropical
*Erythrina lysistemon* Hutch.		X		Tree	LC	Afrotropical—Afrotemperate
*Erythrina speciosa* Andrews	X			Tree	NE	Neotropical
*Gleditsia triacanthos* L.	X	X	X	Tree	LC	Nearctic
*Inga edulis* Mart.			X	Tree	LC	Neotropical—Andean
*Leucaena leucocephala* (Lam.) de Wit		X	X	Tree	NE	Neotropical
*Libidibia paraguariensis* (D.Parodi) G.P.Lewis	X			Tree	VU	Neotropical
*Mimosa aculeaticarpa* Ortega		X		Shrub	LC	Neotropical
*Parkinsonia aculeata* L.	X	X		Tree	LC	Andean—Nearctic—Neotropical
*Peltophorum dubium* (Spreng.) Taub.			X	Tree	LC	Neotropical
*Prosopis chilensis* (Molina) Stuntz	X			Tree	LC	Andean—Neotropical
*Prosopis glandulosa* Torr.	X			Tree	LC	Nearctic—Neotropical
*Prosopis juliflora* (Sw.) DC.			X	Tree	NE	Neotropical
*Prosopis laevigata* (Humb. and Bonpl. ex Willd.) M.C.Johnst.			X	Tree	LC	Neotropical
*Robinia neomexicana* A.Gray var. *neomexicana*		X		Tree	NE	Nearctic—Neotropical
*Robinia pseudoacacia* L.	X	X	X	Tree	LC	Nearctic
*Schotia afra* (L.) Thunb.	X			Tree	LC	Afrotropical—Afrotemperate
*Schotia brachypetala* Sond.		X		Tree	LC	Afrotropical—Afrotemperate
*Schotia latifolia* Jacq.	X		X	Tree	LC	Afrotropical—Afrotemperate
*Senna didymobotrya* (Fresen.) H.S. Irwin and Barneby		X	X	Shrub or Tree	LC	Afrotropical
*Sophora davidi* (Franch.) Skeels		X		Shrub	NE	Oriental
*Sophora microphylla* Aiton	X			Shrub or Tree	NE	Neozelandic
*Styphnolobium japonicum* (L.) Schott	X	X	X	Tree	NE	Oriental
*Tara spinosa* (Molina) Britton and Rose	X	X		Tree	NE	Andean—Neotropical
*Tipuana tipu* (Benth.) Kuntze		X	X	Tree	LC	Neotropical
*Vachellia farnesiana* (L.) Wight and Arn.	X			Tree	LC	Neotropical
*Vachellia karroo* (Hayne) Banfi and Galasso		X	X	Tree	NE	Afrotropical—Afrotemperate
Fagaceae						
*Fagus sylvatica* L.		X		Tree	LC	Palearctic
*Quercus alnifolia* Poech		X		Tree	LC	Palearctic
*Quercus coccifera* L.	X	X		Shrub	LC	Palearctic
*Quercus faginea* Lam. subsp. *broteroi* (Cout.) A.Camus	X			Tree	NE	Palearctic
*Quercus faginea* Lam. subsp. *faginea*	X	X		Tree	LC	Palearctic
*Quercus ilex* L.		X		Tree	LC	Palearctic
*Quercus libani* G.Olivier		X		Tree	LC	Palearctic
*Quercus rotundifolia* Lam.	X		X	Tree	LC	Palearctic
*Quercus suber* L.	X	X	X	Tree	LC	Palearctic
Ginkoaceae						
*Ginkgo biloba* L.	X	X	X	Tree	EN	Oriental
Hamamelidaceae						
*Parrotia persica* C.A.Mey.		X		Tree	NE	Palearctic
Hydrophyllaceae						
*Wigandia urens* (Ruiz and Pav.) Kunth		X		Shrub	LC	Neotropical
Juglandaceae						
*Carya illinoinensis* (Wangenh.) K.Koch		X	X	Tree	LC	Nearctic
*Juglans nigra* L.		X	X	Tree	LC	Nearctic
*Juglans regia* L.		X		Tree	LC	Palearctic
*Pterocarya fraxinifolia* (Poir.) Spach	X			Tree	VU	Palearctic
Lamiaceae						
*Vitex agnus-castus* L.	X	X	X	Tree	DD	Palearctic
Lauraceae						
*Cinnamomum burmanni* (Nees and T.Nees) Blume	X		X	Tree	NE	Oriental
*Cinnamomum camphora* (L.) J.Presl	X	X	X	Tree	NE	Palearctic
*Cinnamomum tamala* (Buch.-Ham.) T.Nees and Eberm.		X		Tree	LC	Oriental
*Cinnamomum verum* J.Presl	X			Tree	NE	Oriental
*Laurus azorica* (Seub.) Franco			X	Tree	LC	Palearctic
*Laurus nobilis* L.	X	X	X	Tree	LC	Palearctic
*Ocotea foetens* (Aiton) Baill.	X	X	X	Tree	LC	Afrotropical
*Persea americana* Mill.	X	X	X	Tree	LC	Neotropical
*Persea barbujana* (Cav.) Mabb. and Nieto Fel.	X	X	X	Tree	LC	Afrotropical
*Persea indica* (L.) Spreng.	X	X	X	Tree	LC	Afrotropical
Lythraceae						
*Lagerstroemia indica* L.	X	X	X	Tree	LC	Oriental
*Punica granatum* L.	X	X	X	Shrub	LC	Palearctic
Magnoliaceae						
*Liriodendron tulipifera* L.		X		Tree	LC	Nearctic
*Magnolia champaca* (L.) Baill. ex Pierre	X			Tree	LC	Oriental
*Magnolia compressa* Maxim.		X		Tree	DD	Oriental
*Magnolia grandiflora* L.	X	X		Tree	LC	Nearctic
*Magnolia tripetala* (L.) L.		X		Tree	LC	Nearctic
*Magnolia**× soulangeana* Soul.-Bod.		X		Tree	NE	Hybrid
Malvaceae						
*Brachychiton acerifolius* (A.Cunn. ex G.Don) F.Muell.	X	X	X	Tree	NE	Australotemperate
*Brachychiton bidwillii* Hook.			X	Tree	NE	Australotemperate
*Brachychiton discolor* F. Muell.	X		X	Tree	NE	Australotemperate
*Brachychiton populneus* (Schott and Endl.) R.Br.	X	X	X	Tree	NE	Australotemperate
*Brachychiton rupestris* (T.Mitch. ex Lindl.) K.Schum.			X	Tree	NE	Australotemperate
*Ceiba crispiflora* (Kunth) Ravenna		X		Tree	NE	Neotropical
*Ceiba insignis* (Kunth) P.E.Gibbs and Semir		X	X	Tree	NE	Andean
*Ceiba pentandra* (L.) Gaertn.			X	Tree	LC	Neotropical
*Ceiba speciosa* (A.St.-Hil., A.Juss. and Cambess.) Ravenna	X	X	X	Tree	NE	Neotropical
*Dombeya burgessiae* Gerrard ex Harv.		X	X	Shrub	LC	Afrotropical
*Dombeya rotundifolia* (Hochst.) Planch.		X		Shrub or Tree	LC	Afrotropical—Afrotemperate
*Dombeya tiliacea* (Endl.) Planch.			X	Tree	LC	Afrotropical—Afrotemperate
*Dombeya* × *cayeuxii* André	X	X	X	Tree	NE	Hybrid
*Firmiana simplex* (L.) W.Wight	X	X		Tree	LC	Oriental
*Grewia occidentalis* L.		X		Tree	LC	Afrotropical—Afrotemperate
*Hibiscus mutabilis* L.	X	X	X	Shrub	NE	Oriental
*Hibiscus rosa-sinensis* L.	X	X	X	Shrub	NE	Oriental
*Hibiscus syriacus* L.	X	X	X	Shrub or Tree	NE	Oriental
*Lagunaria patersonia* (Andrews) G.Don	X	X		Tree	NE	Australotemperate
*Malvaviscus arboreus* Dill. ex Cav.	X		X	Shrub	LC	Neotropical
*Pachira aquatica* Aubl.			X	Tree	LC	Neotropical
*Phymosia umbellata* (Cav.) Kearney	X		X	Shrub or Tree	LC	Neotropical
*Tilia* × *moltkei* Späth ex C.K.Schneid.	X			Tree	NE	Hybrid
*Tilia dasystyla* Steven		X		Tree	LC	Palearctic
*Tilia platyphyllos* Scop.		X		Tree	LC	Palearctic
*Tilia tomentosa* Moench			X	Tree	LC	Palearctic
Meliaceae						
*Cedrela odorata* L.	X		X	Tree	VU	Neotropical
*Melia azedarach* L.	X	X	X	Tree	LC	Australotemperate—Australotropical—Neioguinean—Oriental
*Trichilia emetica* Vahl			X	Tree	LC	Afrotropical
*Trichilia havanensis* Jacq.		X		Tree	LC	Neotropical
Menispermaceae						
*Cocculus laurifolius* DC.		X		Shrub or Tree	NE	Oriental
Monimaceae						
*Peumus boldus* Molina		X		Tree	LC	Andean
Moraceae						
*Broussonetia papyrifera* (L.) L’Hér. ex Vent.	X		X	Tree	LC	Oriental—Palearctic
*Ficus altissima* Blume			X	Tree	LC	Oriental
*Ficus aurea* Nutt.		X		Tree	NE	Neotropical
*Ficus benjamina* L.	X	X	X	Tree	LC	Australotemperate—Oriental
*Ficus carica* L.	X	X	X	Tree	LC	Palearctic
*Ficus coronata* Spin		X		Tree	NE	Australotemperate
*Ficus elastica* Roxb. ex Hornem.	X		X	Tree	NE	Oriental
*Ficus eximia* Schott	X			Tree	LC	Neotropical
*Ficus habrophylla* G.Benn. ex Seem.		X		Tree	NE	Australotemperate
*Ficus luschnathiana* (Miq.) Miq.		X		Tree	LC	Neotropical
*Ficus lyrata* Warb.			X	Tree	NE	Afrotropical
*Ficus macrophylla* Desf. ex Pers.	X	X	X	Tree	NE	Australotemperate
*Ficus microcarpa* L.f.	X		X	Tree	LC	Australotemperate—Neoguinean—Oriental
*Ficus religiosa* L.	X	X	X	Tree	NE	Oriental
*Ficus rubiginosa* Desf. ex Vent.		X		Tree	NE	Australotemperate
*Ficus rumphii* Blume			X	Tree	NE	Oriental
*Ficus superba* (Miq.) Miq.		X		Tree	NE	Oriental
*Ficus sur* Forssk.		X		Tree	LC	Afrotropical
*Ficus sycomorus* L.		X	X	Tree	LC	Afrotropical
*Ficus virens* Aiton			X	Tree	LC	Australotemperate—Australotropical—Oriental
*Maclura pomifera* (Raf.) C.K.Schneid.		X	X	Tree	LC	Nearctic
*Morus alba* L.	X	X	X	Tree	NE	Oriental
*Morus nigra* L.	X			Tree	NE	Palearctic
Myricaceae						
*Myrica faya* Dryand.	X	X		Shrub or Tree	LC	Palearctic
Myrtaceae						
*Agonis flexuosa* (Willd.) Sweet		X		Tree	NE	Australotropical
*Corymbia maculata* (Hook.) K.D.Hill and L.A.S.Johnson			X	Tree	NE	Australotemperate
*Eucalyptus camaldulensis* Dehnh.	X	X	X	Tree	NT	Australotropical—Australotemperate
*Eucalyptus cladocalyx* F.Muell.		X		Tree	VU	Australotemperate
*Eucalyptus cornuta* Labill.		X		Tree	NT	Australotemperate
*Eucalyptus diversicolor* F.Muell.	X			Tree	LC	Australotemperate
*Eucalyptus globulus* Labill.			X	Tree	LC	Australotemperate
*Eucalyptus gomphocephala* A.Cunn ex DC.		X	X	Tree	VU	Australotemperate
*Eucalyptus ovata* Labill.			X	Tree	VU	Australotemperate
*Eucalyptus robusta* Sm.			X	Tree	NT	Australotemperate
*Eucalyptus tereticornis* Sm.		X	X	Tree	LC	Australotemperate
*Eucalyptus* × *kirtoniana* F.Muell.			X	Tree	NE	Australotemperate
*Eugenia involucrata* DC.	X			Shrub	LC	Neotropical
*Eugenia myrcianthes* Nied.			X	Tree	LC	Neotropical
*Eugenia uniflora* L.	X	X		Tree	NE	Neotropical
*Feijoa sellowiana* (O.Berg) O.Berg		X	X	Shrub or Tree	LC	Neotropical
*Leptospermum laevigatum* (Gaertn.) F.Muell.		X		Tree	NE	Australotemperate
*Lophostemon confertus* (R.Br.) Peter G.Wilson and J.T. Waterh.		X	X	Tree	NE	Australotemperate
*Melaleuca armillaris* (Sol. ex Gaertn.) Sm.	X		X	Tree	NE	Australotemperate
*Melaleuca citrina* (Curtis) Dum.Cours.	X			Shrub	NE	Australotemperate
*Melaleuca lanceolata* Otto			X	Tree	LC	Australotemperate
*Melaleuca leucadendra* (L.) L.			X	Tree	NE	Australotropical—Neoguinean
*Melaleuca linearis* Schrad. and J.C.Wendl. var. *linearis*	X	X		Shrub	NE	Australotemperate
*Melaleuca preissiana* Schauer		X		Tree	LC	Australotemperate
*Melaleuca rugulosa* (Link) Craven	X			Shrub	NE	Australotemperate
*Melaleuca styphelioides* Sm.	X			Tree	NE	Australotemperate
*Melaleuca viminalis* (Sol. ex Gaertn.) Byrnes	X			Shrub	NE	Australotemperate
*Melaleuca virens* Craven	X			Shrub	NE	Australotemperate
*Metrosideros excelsa* Sol. ex Gaertn.		X	X	Tree	NE	Neozelandic
*Myrtus communis* L.	X	X	X	Shrub	LC	Afrotropical—Palearctic
*Psidium cattleyanum* Sabine	X	X	X	Shrub or Tree	NE	Neotropical
*Psidium guajava* L.	X		X	Tree	LC	Andean—Neotropical
*Psidium guineense* Sw.			X	Tree	LC	Andean—Neotropical
*Syzygium cumini* (L.) Skeels			X	Tree	LC	Oriental—Australotropical
*Syzygium jambos* (L.) Alston			X	Tree	LC	Oriental
*Syzygium paniculatum* Gaertn.	X	X		Tree	NE	Australotemperate
Ochnaceae						
*Ochna serrulata* (Hochst.) Walp.		X	X	Shrub	LC	Afrotropical—Afrotemperate
Oleaceae						
*Fraxinus angustifolia* Vahl subsp. *angustifolia*	X	X	X	Tree	LC	Palearctic
*Fraxinus anomala* Torr. ex S. Watson		X		Tree	LC	Nearctic
*Fraxinus floribunda* Wall.		X		Tree	LC	Oriental
*Fraxinus ornus* L.		X		Tree	LC	Palearctic
*Fraxinus pennsylvanica* Marshall	X	X		Tree	CR	Nearctic
*Ligustrum japonicum* Thunb.		X	X	Tree	NE	Oriental
*Ligustrum lucidum* W.T. Aiton	X	X	X	Tree	LC	Oriental
*Ligustrum sinense* Lour.		X	X	Tree	LC	Oriental
*Olea capensis* L.		X		Tree	NE	Afrotropical
*Olea europaea* L. subsp. *europaea*	X	X	X	Tree	DD	Palearctic
*Olea europaea* subsp. *cuspidata* (Wall. and G.Don) Cif.			X	Tree	NE	Afrotropical—Palearctic
*Osmanthus fragrans* Lour.			X	Shrub or Tree	LC	Oriental
*Phillyrea latifolia* L.	X	X	X	Tree	LC	Palearctic
*Picconia azorica* (Tutin) Knobl.	X	X		Tree	LC	Palearctic
*Picconia excelsa* (Aiton) DC.		X		Tree	VU	Afrotropical
*Syringa vulgaris* L.	X	X	X	Shrub	LC	Palearctic
Paulowniaceae						
*Paulownia tomentosa* (Thunb.) Steud.	X	X	X	Tree	NE	Palearctic
Phyllanthaceae						
*Phyllanthus juglandifolius* Willd.			X	Shrub or Tree	NE	Andean—Neotropical
Phytolaccaceae						
*Phytolacca dioica* L.	X	X	X	Tree	NE	Andean—Neotropical
Pinaceae						
*Abies concolor* (Gordon and Glend.) Lindl. ex Hildebr.		X		Tree	LC	Nearctic
*Abies pinsapo* Boiss.		X		Tree	EN	Palearctic
*Cedrus atlantica* (Endl.) Manetti ex Carrière		X	X	Tree	EN	Palearctic
*Cedrus deodara* (Roxb. ex D.Don) G.Don		X	X	Tree	LC	Oriental
*Picea laxa* (Münchh.) Sarg.		X		Tree	LC	Nearctic
*Picea pungens* Engelm.		X		Tree	LC	Nearctic
*Pinus bungeana* Zucc. ex Endl.		X		Tree	LC	Oriental
*Pinus canariensis* C.Sm.		X		Tree	LC	Afrotropical
*Pinus halepensis* Mill.		X		Tree	LC	Palearctic
*Pinus nigra* J.F.Arnold			X	Tree	LC	Palearctic
*Pinus pinea* L.	X	X	X	Tree	LC	Palearctic
*Pinus sylvestris* L.		X		Tree	LC	Palearctic
*Pinus teocote* Schiede ex Schltdl. and Cham.			X	Tree	LC	Palearctic
*Pinus torreyana* Parry ex Carrière		X		Tree	CR	Nearctic
Piperaceae						
*Piper jacquemontianum* Kunth			X	Shrub	NE	Neotropical
Pittosporaceae						
*Pittosporum crassifolium* Banks and Sol. ex A.Cunn.		X		Shrub or Tree	NE	Neozelandic
*Pittosporum tobira* (Thunb.) W.T.Aiton	X	X	X	Shrub or Tree	NE	Oriental—Palearctic
*Pittosporum undulatum* Vent.	X	X	X	Tree	NE	Australotemperate
*Pittosporum viridiflorum* Sims		X		Shrub or Tree	NE	Afrotropical
Platanaceae						
*Platanus* × *hispanica* Mill. ex Münchh.	X	X	X	Tree	NE	Hybrid
Podocarpaceae						
*Afrocarpus mannii* (Hook.f.) C.N.Page	X	X	X	Tree	VU	Afrotropical
*Podocarpus macrophyllus* (Thunb.) Sweet	X			Tree	LC	Oriental
*Podocarpus neriifolius* D.Don		X	X	Tree	LC	Oriental
*Podocarpus totara* G.Benn. ex D.Don		X		Tree	LC	Neozelandic
Primulaceae						
*Myrsine africana* L.		X		Shrub	NE	Afrotropical—Afrotemperate—Oriental
Proteaceae						
*Banksia integrifolia* L.f.	X	X		Tree	LC	Australotemperate
*Grevillea olivacea* A.S.George	X			Shrub	LC	Australotropical
*Grevillea robusta* A.Cunn. ex R.Br.	X	X	X	Tree	LC	Australotemperate
*Grevillea thelemanniana* Hügel ex Lindl.	X			Shrub	NE	Australotropical
*Hakea laurina* R.Br.	X			Shrub or Tree	VU	Australotemperate
*Macadamia integrifolia* Maiden and Betche		X	X	Tree	NE	Australotemperate
*Macadamia ternifolia* F. Muell.		X		Tree	EN	Australotemperate
Quillajaceae						
*Quillaja lancifolia* D.Don		X		Tree	NE	Neotropical
Rhamnaceae						
*Colletia paradoxa* (Spreng.) Escal.		X		Shrub	NE	Neotropical
*Colletia spinosissima* J.F.Gmel.		X		Shrub	LC	Andean—Neotropical
*Fontanesia fortunei* Carrière		X		Shrub or Tree	NE	Oriental
*Frangula azorica* Grubov			X	Tree	LC	Palearctic
*Hovenia dulcis* Thunb.		X		Tree	LC	Oriental
*Noltea africana* (L.) Rchb. f.	X			Shrub or Tree	LC	Afrotemperate
*Paliurus spina-christi* Mill.		X		Shrub	NE	Palearctic
*Pomaderris apetala* Labill.		X		Shrub or Tree	NE	Neozelandic
*Rhamnus alaternus* L.	X	X		Shrub	LC	Palearctic
*Rhamnus cathartica* L.	X	X		Tree	LC	Palearctic
*Ziziphus jujuba* Mill.	X	X	X	Tree	LC	Palearctic
Rosaceae						
*Cotoneaster coriaceus* Franch.	X	X	X	Shrub	LC	Palearctic
*Crataegus azarolus* L.		X		Shrub or Tree	LC	Palearctic
*Crataegus monogyna* Jacq.	X	X		Shrub or Tree	NE	Palearctic
*Eriobotrya japonica* (Thunb.) Lindl.	X		X	Tree	NE	Oriental
*Malus domestica* (Suckow) Borkh.			X	Shrub or Tree	EN	Palearctic
*Phoenix sylvestris* (L.) Roxb.		X	X	Tree	NE	Oriental
*Photinia serratifolia* (Desf.) Kalkman		X	X	Shrub or Tree	NE	Oriental
*Prunus avium* (L.) L.			X	Tree	LC	Palearctic
*Prunus azorica* (hort. ex Mouill) Rivas Mart., Lousã, Fern. Prieto, J.C. Costa and C. Aguiar	X			Tree	NE	Palearctic
*Prunus caroliniana* (Mill.) Aiton		X		Tree	LC	Nearctic
*Prunus cerasifera* Ehrh. subsp. *cerasifera*	X	X	X	Tree	DD	Palearctic
*Prunus laurocerasus* L.		X	X	Tree	LC	Palearctic
*Prunus lusitanica* L.		X	X	Tree	EN	Palearctic
*Prunus persica* (L.) Batsch	X	X		Shrub or Tree	NE	Palearctic
*Prunus × blireana* André	X		X	Tree	NE	Hybrid
*Pyracantha angustifolia* (Franch.) C.K.Schneid.		X		Shrub	LC	Palearctic
*Pyracantha crenulata* (D.Don) M.Roem.			X	Shrub	NE	Oriental
*Pyrus pyrifolia* (Burm. f.) Nakai			X	Tree	NE	Oriental
*Rhaphiolepis indica* (L.) Lindl.	X	X	X	Shrub	NE	Oriental
*Spiraea cantoniensis* Lour.	X	X	X	Shrub	LC	Oriental
*Stranvaesia nussia* (Buch.-Ham. ex D.Don)		X		Tree	NE	Oriental—Palearctic
Rubiaceae						
*Coffea arabica* L.	X		X	Shrub or Tree	EN	Afrotropical
*Coffea racemosa* Lour.			X	Shrub or Tree	NT	Afrotropical
*Coprosma repens* A.Rich.	X	X		Shrub	NE	Neozelandic
*Gardenia thunbergia* Thunb.		X	X	Shrub or Tree	NE	Afrotropical—Afrotemperate
*Rogiera amoena* Planch.		X		Shrub or Tree	NE	Neotropical
*Rogiera backhousii* (Hook.f.) Borhidi		X		Shrub or Tree	NE	Neotropical
Rutaceae						
*Calodendrum capense* (L.f.) Thunb.		X		Tree	LC	Afrotropical—Afrotemperate
*Casimiroa edulis* La Llave			X	Tree	LC	Neotropical
*Citrus* × *aurantium* L.	X	X		Tree	NE	Hybrid
*Citrus glauca* (Lindl.) Burkill		X		Shrub or Tree	NE	Australotemperate
*Citrus trifoliata* L.	X	X		Shrub or Tree	NE	Oriental
*Citrus* × *limon* (L.) Osbeck	X	X		Shrub or Tree	NE	Hybrid
*Pilocarpus pennatifolius* Lem.		X	X	Shrub or Tree	NE	Neotropical
*Zanthoxylum armatum* DC.		X	X	Shrub	LC	Oriental
Salicaceae						
*Dovyalis caffra* (Hook.f. and Harv.) Warb.	X	X	X	Tree	LC	Afrotropical—Afrotemperate
*Populus alba* L.		X	X	Tree	LC	Palearctic
*Populus nigra* L.		X	X	Tree	DD	Palearctic
*Salix* × *pendulina* f. *salamonii* (Carrière) I.V.Belyaeva		X		Shrub or Tree	NE	Hybrid
*Salix atrocinerea* Brot.		X		Shrub or Tree	LC	Palearctic
Sapindaceae						
*Acer campestre* L.	X			Tree	LC	Palearctic
*Acer granatense* Boiss.	X			Tree	LC	Palearctic
*Acer monspessulanum* L.	X	X		Tree	LC	Palearctic
*Acer negundo* L.		X		Tree	LC	Nearctic
*Acer palmatum* Thunb.		X		Tree	LC	Palearctic
*Acer pseudoplatanus* L.	X	X		Tree	LC	Palearctic
*Aesculus glabra* Willd.	X			Tree	LC	Nearctic
*Aesculus hippocastanum* L.		X	X	Tree	VU	Palearctic
*Aesculus × carnea* Zeyh.	X	X	X	Tree	NE	Hybrid
*Dodonaea viscosa* Jacq.	X		X	Shrub or Tree	LC	Afrotropical—Australotemperate—Australotropical—Neoguinean—Neotropical—Neozelandic—Oriental
*Harpullia pendula* Planch. ex F.Muell.		X		Tree	NE	Australotemperate
*Hippobromus pauciflorus* (L.f.) Radlk.		X		Tree	LC	Afrotropical—Afrotemperate
*Koelreuteria bipinnata* Franch.		X	X	Shrub or Tree	NE	Oriental
*Koelreuteria paniculata* Laxm.	X	X		Tree	LC	Oriental—Palearctic
*Sapindus drummondii* Hook. and Arn.		X	X	Tree	NE	Nearctic
*Sapindus mukorossi* Gaertn.		X	X	Tree	NE	Oriental—Palearctic
*Sapindus saponaria* L.		X		Tree	LC	Oriental
Sapotaceae						
*Chrysophyllum imperiale* (Linden ex K.Koch and Fintelm.) Benth. and Hook.f.		X	X	Tree	EN	Neotropical
*Sideroxylon inerme* L.		X		Tree	NE	Afrotropical—Afrotemperate
*Sideroxylon mirmulano* R.Br.		X		Tree	EN	Afrotropical
Scrophulariaceae						
*Myoporum laetum* G.Forst.	X	X		Tree	NE	Neozelandic
Simaroubaceae						
*Ailanthus altissima* (Mill.) Swingle		X	X	Tree	NE	Palearctic
Solanaceae						
*Brugmansia aurea* Lagerh.		X		Shrub	EW	Neotropical
*Cestrum × cultum* Francey		X		Shrub	NE	Hybrid
*Nicotiana glauca* Graham	X			Shrub	NE	Andean—Neotropical
*Solanum crinitum* Lam.		X		Shrub or Tree	NE	Neotropical
Stilbaceae						
*Halleria lucida* L.		X		Shrub or Tree	LC	Afrotropical—Afrotemperate
Tamaricaceae						
*Tamarix africana* Poir.	X	X		Shrub or Tree	LC	Palearctic
*Tamarix gallica* L.	X			Shrub or Tree	LC	Palearctic
*Tamarix parviflora* DC.		X		Shrub or Tree	LC	Palearctic
Taxaceae						
*Cephalotaxus harringtonia* (Knight ex J.Forbes) K.Koch		X	X	Shrub or Tree	LC	Oriental
*Taxus baccata* L.	X	X	X	Tree	LC	Palearctic
*Torreya californica* Torr.		X		Tree	VU	Nearctic
Theaceae						
*Camellia japonica* L.	X	X		Shrub	LC	Palearctic
Ulmaceae						
*Ulmus minor* Mill.	X			Tree	DD	Palearctic
*Zelkova serrata* (Thunb.) Makino	X	X		Tree	NT	Oriental—Palearctic
Urticaceae						
*Myriocarpa longipes* Liebm.		X		Shrub or Tree	NE	Neotropical
Verbenaceae						
*Citharexylum spinosum* L.		X		Shrub or Tree	LC	Neotropical
*Duranta erecta* L.	X	X	X	Shrub or Tree	LC	Nearctic—Neotropical
Viburnaceae						
*Sambucus nigra* L.	X	X		Shrub or Tree	LC	Palearctic
*Viburnum odoratissimum* Ker Gawl.		X		Shrub	LC	Oriental
*Viburnum rhytidophyllum* Hemsl.		X		Shrub	NE	Oriental
*Viburnum tinus* L.	X	X	X	Shrub	LC	Palearctic
Xanthorrhoeaceae						
*Kumara plicatilis* (L.) G.D.Rowley	X			Rosette tree	NE	Afrotemperate

^a^ EW, Extinct in Wild; CR, Critically Endangered; EN, Endangered; VU, Vulnerable; NT, Near Threatened; LC, Least Concern; DD, Data Deficient; NE, Not Evaluated [12].

**Table 2 plants-10-01367-t002:** General characterization of each garden of Lisbon (JBA—Botanical Garden of Ajuda; JBL—Lisbon Botanical Garden; and JBT—Tropical Botanical Garden).

Characteristics	JBA	JBL	JBT
Year of creation	1764	1837	1906
Year of inauguration	1768	1878	1914 *
Location	western Lisbon/Ajuda	Central Lisbon	western Lisbon/Belém
Coordinates (latitude/longitude)	38.706205/−9.199421	38.717429/−9.150306	38.698140/−9.203913
Elevation	70–80 m	37–77 m	10–35 m
Area (green spaces)	3.8 ha	5.6 ha	6.4 ha

* Corresponds to inauguration date in current location, at Belém.

## Data Availability

We confirm that all data are original and provided in Tables and Figures within the article.

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
