# Peer review of "Natural and Historical Heritage of the Lisbon Botanical Gardens: An Integrative Approach with Tree Collections"

_plants, 2021, doi:10.3390/plants10071367_

Round 1
Reviewer 1 Report
The article gives a perspective on these three important botanic gardens in Lisbon, though in my opinion they are more in the line of arboreta, since their main emphasis is on a collection of trees. Detailed comments in the file uploaded. The paper is suitable for publication in Plants pending some major changes. My two major concerns are; i) the flow of the headings that do not appear to follow the standard sequence of Introduction, Materials and Method, Results followed by Discussion and Conclusions. Putting Discussion (3) before Materials and Method (4) is non-standard practice in scientific journals (an issue the Editor must take up), and ii) no curatorial information has been given. What accessions systems are used in the gardens? Electronic data base such as BG-Base? Card index? What labelling information is there of the specimens in exhibition and signage? What mapping system is used to locate specimens in the grounds? Such as BG-Map? AutoCad? Etc. The authors mention education, but do not expand on this. Botanic gardens are recognized by their outreach and extension programs that are sadly missing in this article.

Author Response
Reviewer#1:
The article gives a perspective on these three important botanic gardens in Lisbon, though in my opinion they are more in the line of arboreta, since their main emphasis is on a collection of trees. Detailed comments in the file uploaded. The paper is suitable for publication in Plants pending some major changes. My two major concerns are; i) the flow of the headings that do not appear to follow the standard sequence of Introduction, Materials and Method, Results followed by Discussion and Conclusions. Putting Discussion (3) before Materials and Method (4) is non-standard practice in scientific journals (an issue the Editor must take up), and ii) no curatorial information has been given. What accessions systems are used in the gardens? Electronic data base such as BG-Base? Card index? What labelling information is there of the specimens in exhibition and signage? What mapping system is used to locate specimens in the grounds? Such as BG-Map? AutoCad? Etc. The authors mention education, but do not expand on this. Botanic gardens are recognized by their outreach and extension programs that are sadly missing in this article.
RESPONSE: We would like to thank the Reviewer1 for the careful and detailed reading of this manuscript and the thoughtful comments.
- i) First, we would like to state that according to the Journal's submission guidelines, the Methods section is presented at the end of the manuscript. Therefore, we follow the structure proposed in the submission guidelines of the Plants.
- ii) We have carefully considered this comment and so we have profoundly changed the “4.3. Tree Layer Inventory” sub-section in the Material and Methods, and more details are provided in the new version of the ms.
Also, we recognize that some issues were not sufficiently addressed in the previous version of our study, and in the new version of the ms, we have included a new sub-section “3.2. Outreach and education programs” at the end of the Discussion section.
Following are the detailed suggestions for the Authors:
LINE 60-61 - “The only city in world with 3 botanic gardens”. Not true New York has 3; Brooklyn, Queens & NYBG. Los Angeles has 3 +; South Coast BG, Manhatten Beach BG, S. Mark Taper Life Science BG plus three more. San Francisco has 3 BGs, Melrose Detroit BG, San Francisco BG, Succulents Garden. Seattle has 3+ BGs, Carl S. English BG, Bellevue BG, E. Cary Miller BG, and 3 others. Moscow has 3+ Botanicheskiy Sad BG, N.V. Tsitsin BG, Jardín de los Boticarios and four others. Tokyo has 3+ Akatsuka BG, Koishikawa BG, Shissei BG and three others. Among other cities. Perhaps the authors meant the only city in Portugal with three botanic gardens.
RESPONSE: We would like to state that we are aware that several cities in the world have three Botanic Gardens, but in the previous version of our ms we would like to highlight that Lisbon is the only city that holds one University with three Botanical Gardens – University of Lisbon. However, and in order to address the reviewer concern, we corrected the text as suggested by the reviewer, and now reads “Lisbon is the only city in Portugal housing three botanical gardens, and they hold a rich natural and historical heritage, a valuable scientific resource to promote interdisciplinary research activities between the different schools of the University, as well as with worldwide Institutions interested in the study of plant diversity and conservation”.
METHODS: Although an inventory is given in Table 1 and location of each garden given, how was the information gathered? By questionnaire? Literature search? Visits? Etc please expand on this.
RESPONSE: We agree with the Reviewer that this aspect was not sufficiently explained in the first version of our manuscript. Therefore, we added new text in the sub-section 4.3. Tree Layer Inventory: “The surveys in the three botanical gardens focused on tree specimens and were carried out in the framework of the “LX GARDENS” research project (2014-2017, targeting Lisbon’s historic gardens). The following methodology was used for the botanical study: inventory, location (with geographic information systems) and specimen identification. All data were recorded in a relational database built on a SQL server. Location data recorded, included the following items: (1) Specimen ID number; (2) Garden code; (3) Species code; (4) Family; (5) Species; (6) Species classifier; (7) Geo-graphic origin of the taxa; (8) Naturality Status in Portugal (i.e. native non-native and/or invasive); (9) Growth form (meaning the plant´s physiognomy); (10) ETRS 1989 coordinates; (11) Extinction risk assessment using the IUCN Red List [12]. The data of the three botanical gardens were continuously updated until May 2021”.
Line 138 - Change hold to held.
RESPONSE: Done
Figs 2 & 3 - Areas labelled as green appear blue, please check.
RESPONSE: We understand this comment and thus the Fig. 2 legend has been revised, taking into account the Reviewer’s concerns.
Line 279 - Ginkgo biloba should be in italics to read Ginkgo biloba
RESPONSE: Done.
Line 355 - The section Materials and Method should go before Results and Discussion sections. Editor opinion?
RESPONSE: We would like to state that according to the Journal's submission guidelines, the Methods section is presented at the end of the manuscript. Therefore, we follow the structure proposed in the submission guidelines of the Plants.
Line 360 Table 2 - Should not year of inauguration (open to the public) be more informative than year of construction? Perhaps both dates will be informative; year when work on the garden is initiated leading to year of inauguration, when open to the public.
RESPONSE: Following this suggestion, the authors included in the Table 2 detailed information about the year of inauguration, and when was open to the public.
General comment: Botanic gardens are recognized for their extension and outreach programs. Educational role has been mentioned in the discussion. It would be good to expand on this role for each garden. What extension and outreach programs do they offer? How many visitors per year does each garden record? Are there guided tours offered to schools or the public? Other services to the community such as plant identification?
RESPONSE: As mentioned above, we recognize that some issues were not sufficiently addressed in the previous version of our study, and in the new version of the ms, we have included a new sub-section “3.2. Outreach and education programs” at the end of the Discussion section.
Reviewer 2 Report
This delightful paper is an excellent study of the history and importance of the 3 botanic gardens in Lisbon, using the living tree collections of these gardens as a primary measure. It is well organized and well written, and uses a solid methodology. After these few minor items below are addressed, this paper should be published right away.
Line 60-61: "Lisbon is the only city in the world housing three botanical gardens" – Please verify this, or edit the statement. These two web tools might be useful:
https://tools.bgci.org/map.php
https://www.publicgardens.org/about-public-gardens/gardens
Line 74: "the presence of three botanical gardens in Lisbon is not redundant" -- I agree, in fact, the three gardens complement one another and create a collaborative network that is greater than the sum of its parts. I think, though, that having multiple site 'redundancy' is actually very helpful in the case of disasters, as mentioned in line 177.
Line 124: perhaps "succulent dahlias"
Line 125: please use the multiplication symbol "×" for the hybrid.
Line 144: please remove the apostrophe from "1940s"
Line 163: the fact that 85 taxa are shared by all three gardens is actually very good -- especially in light of the potential for hurricanes (or other problems) as mentioned in line 177.
Figure 3: when printing this on paper, the grey colors were kind of difficult to tell apart.
Table 1: on the header of this table (line 223 or so) please indicate the date of the survey. I saw this mentioned later (line 395 or so), but this table is central to the paper, and I think offers an important 'snapshot in time' -- it would be nice if the reader in future years knows exactly when this survey was completed. Thinking ahead, this will be of interest in a decade or two, or in 100 years, when someone can compare this record to what collections are held at that time. Garden tree collections change over time (sometimes rapidly).
Line 254-255: I really appreciated this sentence about Araucaria, and I think more such content discussing the cultural and symbolic value of certain taxa would be nice to include throughout. This helps to provide insight on how the collections were developed also.
Line 302: please change "permanency" to "persistence"
Glad to see the term "tree" defined on line 400.
I also would like to see more illustrations of the various aspects discussed around lines 226-250 -- please add a few more figures that go along with the discussion. For example, the Dracaena, the importance of Araucaria in local horticulture, and the main avenue flanked by Washingtonia would be evocative figures. There is so much about botanic gardens that is hard to describe in words, and photographs are the next best thing. (If you can't be there in person!)
I also think a section about “future directions” would be nice. A great thing about this organized study and comparison of these collections is that it highlights current strengths, but it also suggests areas for future collections development. Some brief material about how this information is begin used to inform future plans would be nice to include.
Again, an excellent study, and I believe it can inspire other gardens to consider their ‘collective collections’ holdings and how this can inform their work.
Author Response
Reviewer#2:
This delightful paper is an excellent study of the history and importance of the 3 botanic gardens in Lisbon, using the living tree collections of these gardens as a primary measure. It is well organized and well written, and uses a solid methodology. After these few minor items below are addressed, this paper should be published right away.
RESPONSE: The authors would like to thank the Reviewers#2 for his very thorough and insightful comments, which have greatly contributed to improving the clarity and readability of our paper.
Line 60-61: "Lisbon is the only city in the world housing three botanical gardens" – Please verify this, or edit the statement. These two web tools might be useful: https://tools.bgci.org/map.php; https://www.publicgardens.org/about-public-gardens/gardens;
RESPONSE: As mentioned above, we would like to state that we are aware that several cities in the world have three Botanic Gardens, but in the previous version of our ms we would like to highlight that Lisbon is the only city that holds one University with three Botanical Gardens – University of Lisbon. However, and in order to address the reviewer concern, we corrected the text as suggested by the reviewer, and now reads “Lisbon is the only city in Portugal housing three botanical gardens, and they hold a rich natural and historical heritage, a valuable scientific resource to promote interdisciplinary research activities between the different schools of the University, as well as with worldwide Institutions interested in the study of plant diversity and conservation”.
Line 74: "the presence of three botanical gardens in Lisbon is not redundant" -- I agree, in fact, the three gardens complement one another and create a collaborative network that is greater than the sum of its parts. I think, though, that having multiple site 'redundancy' is actually very helpful in the case of disasters, as mentioned in line 177.
RESPONSE: Thank you very much for this positive feedback.
Line 124: perhaps "succulent dahlias"
RESPONSE: corrected in the text as suggested by the reviewer.
Line 125: please use the multiplication symbol "×" for the hybrid.
RESPONSE: corrected in the text as suggested by the reviewer.
Line 144: please remove the apostrophe from "1940s"
RESPONSE: corrected in the text as suggested by the reviewer.
Line 163: the fact that 85 taxa are shared by all three gardens is actually very good -- especially in light of the potential for hurricanes (or other problems) as mentioned in line 177.
RESPONSE: Thank you very much for this positive feedback.
Figure 3: when printing this on paper, the grey colors were kind of difficult to tell apart.
RESPONSE: The quality of this Figure was improved as suggested by the reviewer, but we prefer to keep the same colors as this pattern is followed for the figures 2, 3 and 5.
Table 1: on the header of this table (line 223 or so) please indicate the date of the survey. I saw this mentioned later (line 395 or so), but this table is central to the paper, and I think offers an important 'snapshot in time' -- it would be nice if the reader in future years knows exactly when this survey was completed. Thinking ahead, this will be of interest in a decade or two, or in 100 years, when someone can compare this record to what collections are held at that time. Garden tree collections change over time (sometimes rapidly).
RESPONSE: corrected in the text as suggested by the reviewer.
Line 254-255: I really appreciated this sentence about Araucaria, and I think more such content discussing the cultural and symbolic value of certain taxa would be nice to include throughout. This helps to provide insight on how the collections were developed also.
RESPONSE: We understand this comment, and new information has been included in the text. It now reads: Particularly, Araucaria spp. are majestic trees very appreciated in Portuguese gardens, with the emblematic examples of A. angustifolia, A. bidwilli, A. cunninghamii, and A. heterophylla standing out. Of these, A. angustifolia and A. heterophylla are in risk of extinction and are therefore categorized as CR and VU, respectively (see Table 1). Although, in the South American rainforests, A. angustifolia have been cultivated since ancient times [34] and indigenous population use their seeds for food and religious rituals [35], the Araucaria forests had their integrity compromised due to overexploitation of timber, aggravated by deforestation for agriculture and urbanism. Due to this fact, a large part of the area covered by A. angustifolia in their native range was eliminated, remaining only a residual part of about 15% [36].
Line 302: please change "permanency" to "persistence"
RESPONSE: corrected in the text as suggested by the reviewer.
Glad to see the term "tree" defined on line 400.
I also would like to see more illustrations of the various aspects discussed around lines 226-250 -- please add a few more figures that go along with the discussion. For example, the Dracaena, the importance of Araucaria in local horticulture, and the main avenue flanked by Washingtonia would be evocative figures. There is so much about botanic gardens that is hard to describe in words, and photographs are the next best thing. (If you can't be there in person!)
RESPONSE: We have carefully considered this comment by the Reviewer, and so we have included three different figures “Figs. 6, 7 and 8”.
I also think a section about “future directions” would be nice. A great thing about this organized study and comparison of these collections is that it highlights current strengths, but it also suggests areas for future collections development. Some brief material about how this information is begin used to inform future plans would be nice to include.
RESPONSE: We have carefully considered this comment and so we have profoundly changed the manuscript. Specifically, in the revised version of the manuscript new information is provided in the two new sub-sections “3.2. Outreach and education programs” and “3.3. Final Remarks” of the Discussion section.
Again, an excellent study, and I believe it can inspire other gardens to consider their ‘collective collections’ holdings and how this can inform their work.
RESPONSE: We would like to thank the Reviewer#2 for the careful and detailed reading of this manuscript, the thoughtful comments, and the positive feedback.
Reviewer 3 Report
Dear authors,
This manuscript constitutes a very interesting and well performed study concerning natural and historical heritage of the Lisbon Botanical Gardens. The overall idea is very interesting, and the methodology is well described. The manuscript, in general, is clear and well-structured as also the interpretation and the discussion of the results. In my opinion the introduction could be enriched with references also focusing on the importance of Botanical gardens. The results presented in 2.2.1-2.2.3 should also be enriched with more information for all the taxa that are mentioned as remarkable (for some of the taxa it is not explained why they are mentioned as remarkable). Additionally, all taxa of conservation importance (as are mentioned in 2.2.2) should be remarked in a Table with more information about their conservation importance. More emphasis should also be given in Discussion to highlight the natural and historical heritage of the Lisbon Botanical Gardens.
Please see among other references:
https://www.sciencedirect.com/science/article/pii/S1878029615002674
https://www.tandfonline.com/doi/full/10.1080/00207233.2017.1284383?casa_token=OwCRRDuqx1gAAAAA%3ASlgcrj4fYDKQhGZGfVExTkZfViGddbXhL6EPadDYoge7X4wAmFI82jKfS3kFeEfZuGOniWOJoofA
https://www.sciencedirect.com/science/article/pii/S1618866717302480?casa_token=TFZB1NPiyzoAAAAA:W0PHhbGCgSqH-RrqRxfjApCwGYMj1uRfeYTuBEgYP4WJTNNNR5TNlRZURvmZwVrssKjsDPWZNw
Author Response
Reviewer#3:
Following are the detailed suggestions for the Authors:
This manuscript constitutes a very interesting and well performed study concerning natural and historical heritage of the Lisbon Botanical Gardens. The overall idea is very interesting, and the methodology is well described. The manuscript, in general, is clear and well-structured as also the interpretation and the discussion of the results. In my opinion the introduction could be enriched with references also focusing on the importance of Botanical gardens.
RESPONSE: Thank you very much for this positive feedback. As suggested, we added new references, including the ones suggested by the reviewer and others to support all the new information added to the revised version of our ms, namely:
Cavender, N.; Smith, P.; Marfleet, K. The Role of Botanic Gardens in Urban Greening and Conserving Urban Biodiversity; Botanic Gardens Conservation International: U.K., Richmond, 2019; p. 25.
Cibrian, A.; Hird Meyer, A.; Oleas, N.; Ma, H.; Meerow, A.; Francisco-Ortega, J.; Griffith, M. What Is the Conservation Value of a Plant in a Botanic Garden? Using Indicators to Improve Management of Ex Situ Collections. The Botanical Review 2013, 79(4), 559-577, doi:10.1007/s12229-013-9120-0.
Hotimah, O.; Wirutomo, P.; Alikodra, H. Conservation of World Heritage Botanical Garden in an Environmentally Friendly City. Procedia Environmental Sciences 2015, 28, 453–463, doi:10.1016/j.proenv.2015.07.055.
Nath, C.; Aravajy, S.; Razasekaran, D.; G, M. Heritage Conservation and Environmental Threats at the 192-Year-Old Botanical Garden in Pondicherry, India. Urban Forestry & Urban Greening 2018, 31, 241–251, doi:10.1016/j.ufug.2018.02.004.
Oldfield, S.F. Botanic Gardens and the Conservation of Tree Species. Trends in Plant Science 2009, 14(11), 581–583, doi:10.1016/j.tplants.2009.08.013.
Qumsiyeh, M.; Handal, E.; Chang, J.; Abualia, K.; Najajrah, M.; Abusarhan, M. Role of Museums and Botanical Gardens in Ecosystem Services in Developing Countries: Case Study and Outlook. International Journal of Environmental Studies 2017, 74(2), 340–350, doi:10.1080/00207233.2017.1284383.
The results presented in 2.2.1-2.2.3 should also be enriched with more information for all the taxa that are mentioned as remarkable (for some of the taxa it is not explained why they are mentioned as remarkable). Additionally, all taxa of conservation importance (as are mentioned in 2.2.2) should be remarked in a Table with more information about their conservation importance.
RESPONSE: We recognize that this was an important remark, and we would like to clarify that in the new version of the ms we provide the conservation status for all the assessed species (please see a new column in the Table 1), and a new Figure (Figure 4) was added including data of IUCN classification of the taxa composing the arboreal stratum of the three Botanical Gardens of Lisbon. Moreover, we have made changes to the text accordingly, namely in the Results and Discussion sections.
More emphasis should also be given in Discussion to highlight the natural and historical heritage of the Lisbon Botanical Gardens.
RESPONSE: We have carefully considered this comment and so we have profoundly changed the Discussion section, and three sub-sections (3.1. Natural and historical heritage of the Botanical Gardens of Lisbon; 3.2. Outreach and education programs and 3.3. Final Remarks) were considerable improved.
Please see among other references:
https://www.sciencedirect.com/science/article/pii/S1878029615002674
https://www.tandfonline.com/doi/full/10.1080/00207233.2017.1284383?casa_token=OwCRRDuqx1gAAAAA%3ASlgcrj4fYDKQhGZGfVExTkZfViGddbXhL6EPadDYoge7X4wAmFI82jKfS3kFeEfZuGOniWOJoofA
https://www.sciencedirect.com/science/article/pii/S1618866717302480?casa_token=TFZB1NPiyzoAAAAA:W0PHhbGCgSqH-RrqRxfjApCwGYMj1uRfeYTuBEgYP4WJTNNNR5TNlRZURvmZwVrssKjsDPWZNw
RESPONSE: We agree with the reviewer and we have included these references in the introduction and Discussion sections.
Finally, all the language in the manuscript has been extensively checked and revised by a native speaker and senior researcher, who has improved the syntax and the text’s general fluidity. The suggestions put forward, by the Reviwers were taken into account in the review process.
Once more, the authors would like to thank the Reviewers for their detailed and insightful comments, and we hope that these improvements have adequately addressed all the concerns.
We look forward to hearing from you at your earliest convenience.
Maria Romeiras, on behalf of all the authors
Round 2
Reviewer 1 Report
Please check all the literature cited in the text is present in literature cited. Run an English spell check for minor typos.
The paper is interesting and well presented and I recommend its publication.
Author Response
The literature cited in the text was checked and the DOI were all included in the new version of the ms. Also, the language in the manuscript has been extensively checked and revised by a native speaker and senior researcher, who has improved the syntax and the text’s general fluidity.
Once more, the authors would like to thank the Reviewer 1 for the detailed and insightful comments, and we hope that these improvements have adequately addressed all the concerns.
We look forward to hearing from you at your earliest convenience.
Maria Romeiras, on behalf of all the authors
Reviewer 3 Report
Dear authors,
I think your manuscript is now much updated and it can be accepted for publication.
Kind regards
Author Response
The literature cited in the text was checked and the DOI were all included in the new version of the ms. Also, the language in the manuscript has been extensively checked and revised by a native speaker and senior researcher, who has improved the syntax and the text’s general fluidity.
Once more, the authors would like to thank the Reviewer 3 for the detailed and insightful comments, and we hope that these improvements have adequately addressed all the concerns.
We look forward to hearing from you at your earliest convenience.
Maria Romeiras, on behalf of all the authors